# Short heat shock factor A2 regulates heat resistance and growth balance in *Arabidopsis*

**Wanxia Chen, Jiaqi Zhao, Zhanxia Tao, Shan Zhang, Xiujuan Bei, Wen Lu, Xiaoting Qi\***

College of Life Sciences, Capital Normal University, Beijing, China

## eLife Assessment

The paper reports **valuable** findings about the mechanism of regulation of the heat shock response in plants that acts as a brake to prevent hyperactivation of the stress response, which have theoretical or practical implications for a subfield. The study presented by the authors provides **solid** methods, data, and analysis that broadly support the claims. This report presents helpful information regarding new spliced HSFs forms in Arabidopsis that highlights key information in the understanding of heat stress and plant growth.

**\*For correspondence:**
qixiaoting@cnu.edu.cn

**Competing interest:** The authors declare that no competing interests exist.

**Abstract** Cells prevent heat damage through the highly conserved canonical heat stress response (HSR), where heat shock factors (HSFs) bind heat shock elements (HSEs) to activate heat shock proteins (HSPs). Plants generate short HSFs (S-HSFs) derived from *HSF* splicing variants, yet their functions remain poorly understood. While an enhanced canonical HSR confers thermotolerance, its hyperactivation inhibits plant growth. How plants prevent this hyperactivation to ensure proper growth remains unknown. Here, we report that *Arabidopsis* S-HsfA2, S-HsfA4c, and S-HsfB1 confer sensitivity to extreme heat (45°C) and constitute new HSF types featuring a unique truncated DNA-binding domain (tDBD). This tDBD binds a new heat-regulated element (HRE), which confers minimal promoter heat-responsiveness and exhibits heat stress sensing and transmission patterns. Using S-HsfA2, we investigated whether and how S-HSFs prevent canonical HSR hyperactivation. *HSP17.6B*, a common direct target of HsfA2 and S-HsfA2, confers thermotolerance; however, its overexpression partially causes HSR hyperactivation. Moreover, HRE–HRE-like and HSE elements mediate the *HSP17.6B* promoter's heat response. We further show S-HsfA2 alleviates hyperactivation via two mechanisms: (1) S-HsfA2 negatively regulates *HSP17.6B* via the HRE–HRE-like element, establishing a noncanonical HSR (S-HsfA2-HRE-*HSP17.6B*) that antagonistically represses HsfA2-activated *HSP17.6B* expression. (2) S-HsfA2 binds the HsfA2 DBD, preventing HsfA2 from binding HSEs and thereby attenuating HsfA2-activated *HSP17.6B* promoter activity. Overall, our findings highlight the essential role of S-HsfA2 in preventing hyperactivation of plant heat tolerance to maintain proper growth.

## Introduction

Elevated temperatures, that is, heat shock or heat stress, have become a global problem that threatens crop yields (*Bita and Gerats, 2013*). Cells prevent heat damage through the canonical heat stress response (HSR). In this canonical HSR, a heat shock factor (HSF) trimer binds a DNA *cis*-acting element called a heat shock element (HSE) to induce the production of heat shock proteins (HSPs). This canonical HSR is highly conserved among different organisms but is highly

complex in plants (*Morimoto, 1993*). Accordingly, heat-tolerant plants are usually generated by overexpressing *HSFs* or *HSPs* (*Fragkostefanakis et al., 2015*). Several studies have revealed that an enhanced canonical HSR is beneficial for plant heat tolerance but also results in growth inhibition under normal conditions. *Zhu et al., 2009* reported that the overexpression of *Boea hygrometrica Hsf1* in *Arabidopsis* and tobacco confers plant thermotolerance but also leads to seedling dwarfism under normal conditions. When *Arabidopsis HsfA2* and *HsfA3* are overexpressed, these *Arabidopsis* plants tolerate increased heat stress but exhibit seedling dwarfism under nonstress conditions (*Ogawa et al., 2007*; *Yoshida et al., 2008*). These findings reflect the active growth inhibition of canonical HSR hyperactivation under nonstress conditions, which is undesirable for plant productivity. However, much less is known about how plants prevent canonical HSR hyperactivation to ensure a balance between heat resistance and growth. Considering that HSRs are highly complex in plants, we reasoned that new types of HSFs that inhibit canonical HSR hyperactivation in plants exist. However, these new types of HSFs and their underlying regulatory mechanisms have yet to be determined in plants.

A common way for introns to expand the diversity of host gene products is alternative splicing (AS). An increasing number of studies have shown that heat stress-induced AS events in heat response genes, including HSFs, are very important regulatory mechanisms for the fitness of plants under heat stress (*Ling et al., 2021*). Thus, we focused on heat-induced AS events in HSFs as a starting point for revealing new kinds of HSFs and their functions and regulatory mechanisms in the plant response to heat stress.

HSFs share similar constructs across eukaryotes, in which the HSE DNA-binding domain (DBD) is the most conserved (*Nover et al., 2001*). The DBD is a 'winged' helix-turn-helix protein that consists of three α-helices (H1, H2, and H3) and four-stranded antiparallel β-sheets (β1, β2, β3, and β4). H3 directly binds to the HSE, and its adjacent wing domain (β3-loop-β4) is needed for the formation of HSF–HSE trimers (*Littlefield and Nelson, 1999*; *Ahn et al., 2001*; *Feng et al., 2021*). Interestingly, for all HSFs, the position of an intron within the DBD is conserved (*Nover et al., 2001*), and the intron is inserted between H3 and the wing domain coding region. In plants, mostly under heat stress, this intron is fully or partially retained to generate splice variants, such as *Arabidopsis HsfA2-II* (*Sugio et al., 2009*), *Arabidopsis HsfA2-III* (*Liu et al., 2013*), rice (*Oryza sativa*) *OsHSFA2dII* (*Cheng et al., 2015*), lily (*Lilium* spp.) *LlHSFA3B-III* (*Wu et al., 2019*), wheat (*Triticum aestivum*) *TaHsfA2-7-AS* (*Ma et al., 2023*), and maize (*Zea mays*) *ZmHsf17-II* (*Zhang et al., 2024*). These variants contain in-frame stop codons within the retained intron that may translate into short HSF isoforms (S-HSFs).

Several S-HSFs have been proven to play regulatory roles in thermotolerance in plants. Previously, we reported that heat stress-induced S-HsfA2 protein was detected by immunoblotting and can regulate the expression of *HsfA2* by binding to HSEs within the *HsfA2* promoter (*Liu et al., 2013*). *Wu et al., 2019* reported that LlHSFA3B-III (S-LlHSFA3B) can be detected by immunoblotting and that overexpressing *S-HSFA3B* reduces tolerance to acute heat (45°C) in *Arabidopsis* plants. TaHsfA2-7 generates the splice variant TaHsfA2-7-AS (S-TaHsfA2-7), which confers tolerance to heat (45°C) in *Arabidopsis* (*Ma et al., 2023*).

S-LlHSFA3B interacts with lily HSFA3A to limit its transactivation function and temper the function of lily HSFA3A (*Wu et al., 2019*). S-ZmHsf17 can interact with the DBD of full-length ZmHsf17 to suppress the transactivation of ZmHsf17 (*Zhang et al., 2024*). These findings suggest that S-HSFs can regulate the activities of HSFs through S-HSF–HSF protein interactions. However, how S-HSFs, as transcription factors, regulate plant heat responses remains unclear. Unlike classical HSFs, S-HSFs contain a unique C-terminal truncated DBD (tDBD) and an extended motif or domain encoded by the retained intron sequences (*Liu et al., 2013*). Although S-HSFs lack nuclear localization signals, S-HSFs can also localize to the nucleus (*Sugio et al., 2009*; *Liu et al., 2013*; *Cheng et al., 2015*; *Wu et al., 2019*; *Ma et al., 2023*; *Zhang et al., 2024*). The tDBD lacks a wing domain and thereby might enable S-HSFs to recognize new heat-responsive DNA elements related to the HSE, whereas extended motifs or domains are variable in length and sequence and subsequently might contribute to regulatory roles, such as transcriptional regulation. Thus, S-HSFs could represent new kinds of plant HSFs. Considering that unique S-HSFs are expressed mostly under extreme heat stress, they could be associated with plant responses to heat stress. Therefore, uncovering the mechanism through which canonical HSR hyperactivation is regulated by S-HSFs will shed light on the balance between growth and thermotolerance.

In this study, we reported that S-HSFs (i.e., S-HsfA2, S-HsfA4c, and S-HsfB1) are new kinds of HSFs that bind new heat-regulated elements (HREs) and negatively regulate *Arabidopsis* tolerance to extreme heat stress. Using S-HsfA2, we further investigated the molecular mechanisms by which S-HsfA2 prevents canonical HSR hyperactivation. The results showed that S-HsfA2 alleviates *HSP17.6B* overexpression-mediated HSR hyperactivation in two different ways: antagonistically repressing *HSP17.6B* overexpression through a noncanonical HSR and acting as a negative binding regulator of HSFs to inhibit the binding of HsfA2 to the HSE of the *HSP17.6B* promoter. Our results reveal new kinds of HSFs originating from HSF splicing variants and provide deep mechanistic insights into proper growth control against thermotolerance hyperactivity in plants.

## Results

### S-HsfA2 confers sensitivity to extreme heat (45°C) sensitivity and inhibits growth in *Arabidopsis*

S-HsfA2 contains a 42-aa N-terminal domain (N-ter), a 61-aa tDBD, and a 26-aa extended leucine (L)-rich domain (LRD) (*Figure 1A*). According to the predicted AlphaFold 3D structure of S-HsfA2, the LRD forms an α-helix structure (*Figure 1A*). Given that S-HsfA2 is expressed under extreme heat (42°C) but not under moderate heat (37°C) (*Liu et al., 2013*), we determined its role in the tolerance of *Arabidopsis* to extreme heat. Compared with the wild-type (WT) control, overexpression of Flag-tagged S-HsfA2 under the control of the cauliflower mosaic virus 35S RNA promoter (35S) (*35S:S-HsfA2-Flag*, S-HsfA2-OE) significantly reduced the seedling survival rate at 45°C for 2 hr (*Figure 1B*).

Our previous study revealed that LRD is responsible for the transcriptional repression of S-HsfA2 in yeast cells (*Liu et al., 2013*). We noted that a conserved transcriptional repression motif, LxLxLx (x = any amino acid) (*Tiwari et al., 2004*), exists in the LRD (*Figure 1A*). Mutations (L to alanine (A)) in the LxLxLx motif (S-HsfA2$^{L-A}$) converted S-HsfA2 from a transcriptional repressor to a transcriptional activator in yeast cells (*Figure 1—figure supplement 1*), indicating that S-HsfA2 is an LxLxLx-type transcriptional repressor. Unlike S-HsfA2-OE, the overexpression of this dominant activator of S-HsfA2-Flag (S-HsfA2$^{L-A}$-OE) failed to reduce the seedling survival rate compared with that of the WT control. However, S-HsfA2$^{L-A}$-OE was more heat tolerant than S-HsfA2-OE was (*Figure 1B*). When the plants were grown in soil, the S-HsfA2 genetic plants also presented a similar extreme heat response phenotype (*Figure 1B*).

Next, we generated S-HsfA2-encoding mRNA (*HsfA2-III*)-knockdown transgenic lines (S-HsfA2-KD) through RNA interference (RNAi) (*Figure 1C*). In three S-HsfA2-KD lines, RT−PCR splicing analysis revealed that *HsfA2-III* is not easily detected. Further RT−qPCR analysis revealed that the abundance of *HsfA2-III* and *HsfA2-II*, but not the full-length *HsfA2* mRNA (except L1#), significantly decreased under extreme heat (42°C). Considering that *HsfA2-III*, but not *HsfA2-II*, is a predominant splice variant under extreme heat (*Liu et al., 2013*), RNAi led to the knockdown of alternative *HsfA2* splicing transcripts, especially *HsfA2-III* in S-HsfA2-KD. S-HsfA2-KD plants presented a clear resistance phenotype to extreme heat in both medium and soil (*Figure 1C*).

We also noted that the overexpression of S-HsfA2 caused short root length (growth in medium) and seedling dwarfism (growth in soil) under normal conditions (*Figure 1D*). However, these growth defects were partially improved in the S-HsfA2$^{L-A}$-OE seedlings (*Figure 1D*).

Overall, these genetic data strongly confirm that S-HsfA2 confers extreme heat (45°C) sensitivity and inhibits growth in *Arabidopsis* seedlings. More importantly, the LxLxLx motif is needed for S-HsfA2 biological functions.

### Comparative transcriptome analysis identifies putative heat stress-responsive genes modulated by S-HsfA2

S-HsfA2$^{L-A}$-OE plants exhibit greater thermotolerance than S-HsfA2-OE controls do, indicating that converting the transcriptional repressor S-HsfA2 into an activated form potentially modulates the expression levels of some heat stress-responsive genes (HRGs). To investigate this hypothesis, we conducted heat (42°C for 1 hr)-induced comparative transcriptome profiling of seedlings of S-HsfA2-OE and S-HsfA2$^{L-A}$-OE (*Figure 2A*). The mRNA sequencing data met the quality control requirements (Q20 >97%, Q30 >92%), confirming that the sequencing quality was sufficient. HRG identification criteria were set as |log$_2$(fold change)| ≥ 1 ($q < 0.05$; $n = 2$ for S-HsfA2-OE, $n = 3$ for

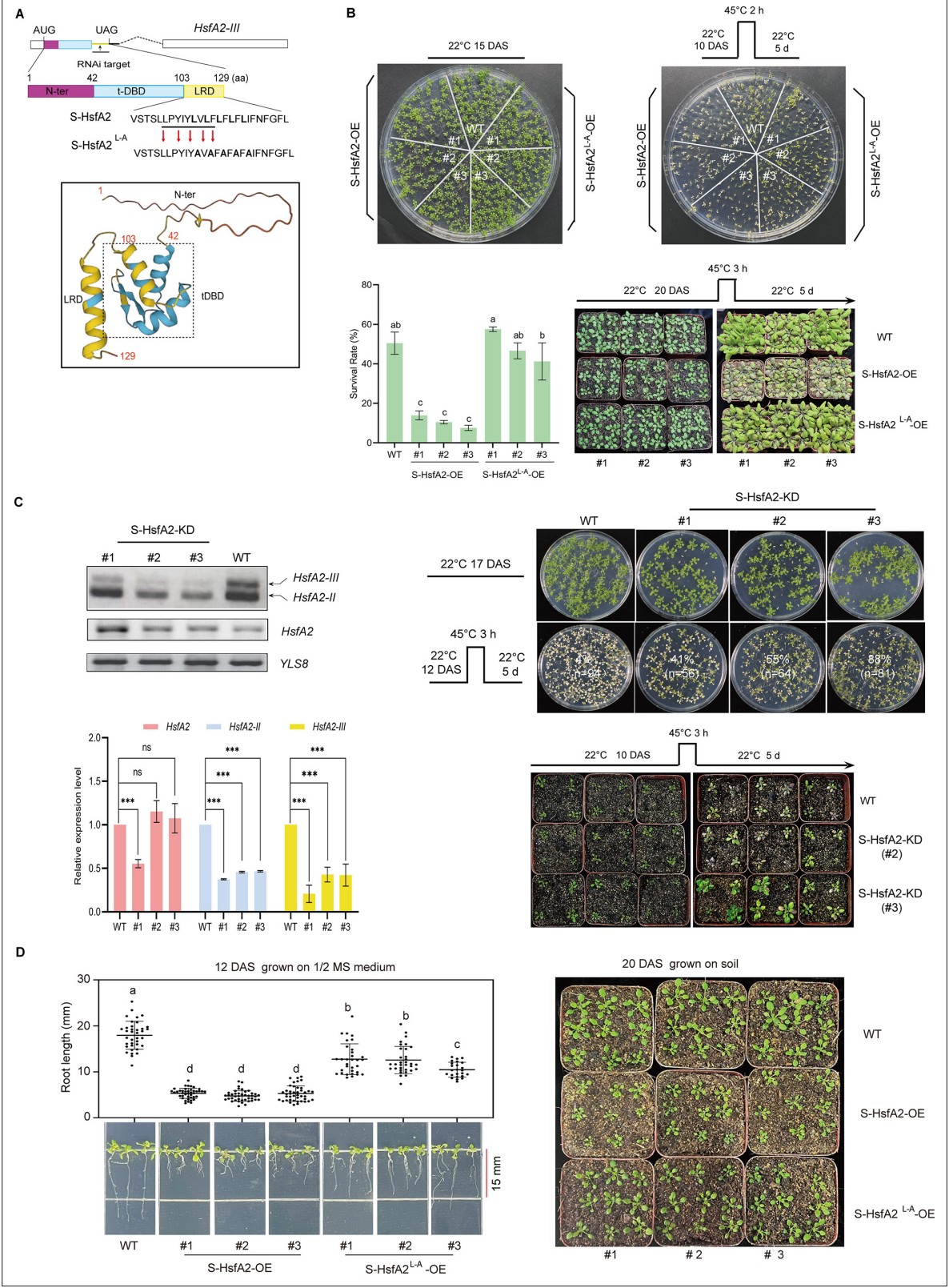

**Figure 1.** S-HsfA2 negatively regulates extreme heat tolerance and growth in *Arabidopsis*. (**A**) Schematic diagrams of *HsfA2-III* and its corresponding product S-HsfA2 (top). Mutations in the LxLxLx motif (underlined) generated the dominant version of S-HsfA2 (S-HsfA2^L-A; see also ***Figure 1—figure supplement 1***). The predicted AlphaFold 3D structure of S-HsfA2 (AFDB accession: AF-J7G1E7-F1) is shown (bottom). (**B**) Thermotolerance of the wild-type (WT) control, S-HsfA2-overexpressing (S-HsfA2-OE) lines, and S-HsfA2^L-A-overexpressing (S-HsfA2^L-A-OE) lines in both medium (top) and

*Figure 1 continued on next page*

*Figure 1 continued*

soil (bottom). A representative image is shown. All green cotyledons were considered surviving, and all yellow cotyledons were considered dead. The survival rates are presented as the means ± SDs of three independent experiments. Different letters indicate significant differences according to one-way ANOVA, $p < 0.05$. (C) RNA interference (RNAi)-mediated S-HsfA2 knockdown (targeting the retained intron sequences; see (A); S-HsfA2-KD) (left) and thermotolerance analyses of S-HsfA2-KD lines (right). Three S-HsfA2-KD lines and the WT control at 12 DAS were treated at 42°C for 1 hr and then subjected to RT−PCR splicing analysis and RT−qPCR analysis. YELLOW-LEAF-SPECIFIC GENE 8 (*YLS8*) served as a loading control. The data are presented as the means ± SDs of two independent experiments. The significance of differences between the experimental values was assessed by Student's *t* test (\*$p < 0.05$). (D) S-HsfA2-OE resulted in reduced root length (left) and a dwarf phenotype (right) in *Arabidopsis* seedlings. Mutations (L to A) in the LxLxLx motif (S-HsfA2$^{L-A}$-OE) partially rescued these growth defects. The data are presented as the means ± SDs of at least two independent experiments. Different letters indicate significant differences according to one-way ANOVA, $p < 0.05$. DAS, days after sowing.

The online version of this article includes the following source data and figure supplement(s) for figure 1:

**Source data 1.** PDF file containing original RT-PCR splicing analysis for *Figure 1C*, showing the relevant bands boxed and labelled.

**Source data 2.** Original files for RT-PCR splicing analysis in *Figure 1C*.

**Figure supplement 1.** The LxLxLx motif is responsible for S-HsfA2 transcriptional repression.

S-HsfA2$^{L-A}$-OE). A total of 1455 HRGs were identified in S-HsfA2-OE (658 upregulated and 797 downregulated; *Figure 2B*; *Supplementary file 1*). In S-HsfA2$^{L-A}$-OE, 1646 HRGs (726 upregulated and 920 downregulated; *Figure 2B*; *Supplementary file 2*) were found. A total of 848 HRGs (58.3% of S-HsfA2-OE HRGs and 51.5% of S-HsfA2$^{L-A}$-OE HRGs; *Figure 2C*; *Supplementary file 3*) overlapped between S-HsfA2$^{L-A}$-OE HRGs and S-HsfA2-OE HRGs. We further analysed the enrichment of Gene Ontology (GO) terms (by ordering p-values and gene numbers; *Figure 2D*; *Supplementary file 4*) among these shared HRGs. GO terms related to the response to heat stress, especially the response to heat, protein folding, and heat acclimation, were identified.

Subsequent comparative analysis between transgenic lines using a $|\log_2(\text{fold change})| \geq 0.5$ threshold identified 352 differentially regulated HRGs (208 upregulated and 144 downregulated in S-HsfA2$^{L-A}$-OE relative to S-HsfA2-OE; *Figure 2C*). Given that converting S-HsfA2 transcriptional repression version (S-HsfA2) to activation version (S-HsfA2$^{L-A}$) can upregulate relative expression levels of shared HRGs, these 208 genes may represent putative S-HsfA2-modulated genes. Interestingly, several heat tolerance-associated genes, including nine *HSPs*, three *HSFs* (*HsfA3*, *HsfA4*, and *HsfA7B*), and three dehydration-responsive element-binding transcription factor genes (*DREBs*), were identified in these 208 HRGs (*Figure 2E*).

## S-HsfA2 binds a *cis*-acting element termed the HRE

As S-HsfA2 is a transcription factor with a unique tDBD, we identified new *cis*-acting elements other than HSEs recognized by S-HsfA2 through chromatin immunoprecipitation combined with high-throughput sequencing (ChIP-Seq). To address this, we generated overexpression lines of GFP-tagged S-HsfA2 under 35S (*35S:S-HsfA2-GFP*). In two independent transgenic lines (#4 and #6), S-HsfA2-GFP signals were hardly detected under normal conditions. However, S-HsfA2-GFP was easily found in the nucleus after heat stress (42°C) treatment (*Figure 3A*). These results suggested that S-HsfA2-GFP accumulates upon heat stress. This result is consistent with the finding that S-HsfA2 is localized in the nucleus of *Arabidopsis* mesophyll protoplasts (*Liu et al., 2013*). Two transgenic lines presented a heat stress-sensitive phenotype (*Figure 3A*), suggesting that S-HsfA2-GFP is biologically functional and may be subjected to ChIP assays.

ChIP-Seq for these two lines with a GFP antibody identified 2163 and 3365 ChIP-Seq peaks (within the ±3 kb region of the transcription start site (TSS)), representing 2103 and 3216 unique genes in #4 and #6, respectively. ChIP-Seq dataset analysis revealed a centrosymmetric 7-bp motif (5′-GAAGAAG-3′). This motif was most significantly enriched ($E = 10^{-19}$) in the ChIP-Seq dataset (*Figure 3B*). Because it was demonstrated to be a heat-regulated element, we hereafter name it an HRE. Since 322 genes were shared between these two lines, we identified 322 putative target genes of S-HsfA2. Given that transcription factors regulate target genes by binding to *cis*-elements within their promoters, the genomic distribution of the peaks revealed 25% enrichment in the promoter-TSS region corresponding to 80 candidate targets of S-HsfA2 (*Figure 3B*, *Supplementary file 5*). Among the 80 candidate targets, 65 contained HREs in their promoters (*Supplementary file 5*).

HSEs are composed of at least three alternating nGAAn/nTTCn blocks (n denotes any nucleotide) (*Amin et al., 1988*). Accordingly, an HRE is considered a partially overlapping element of two

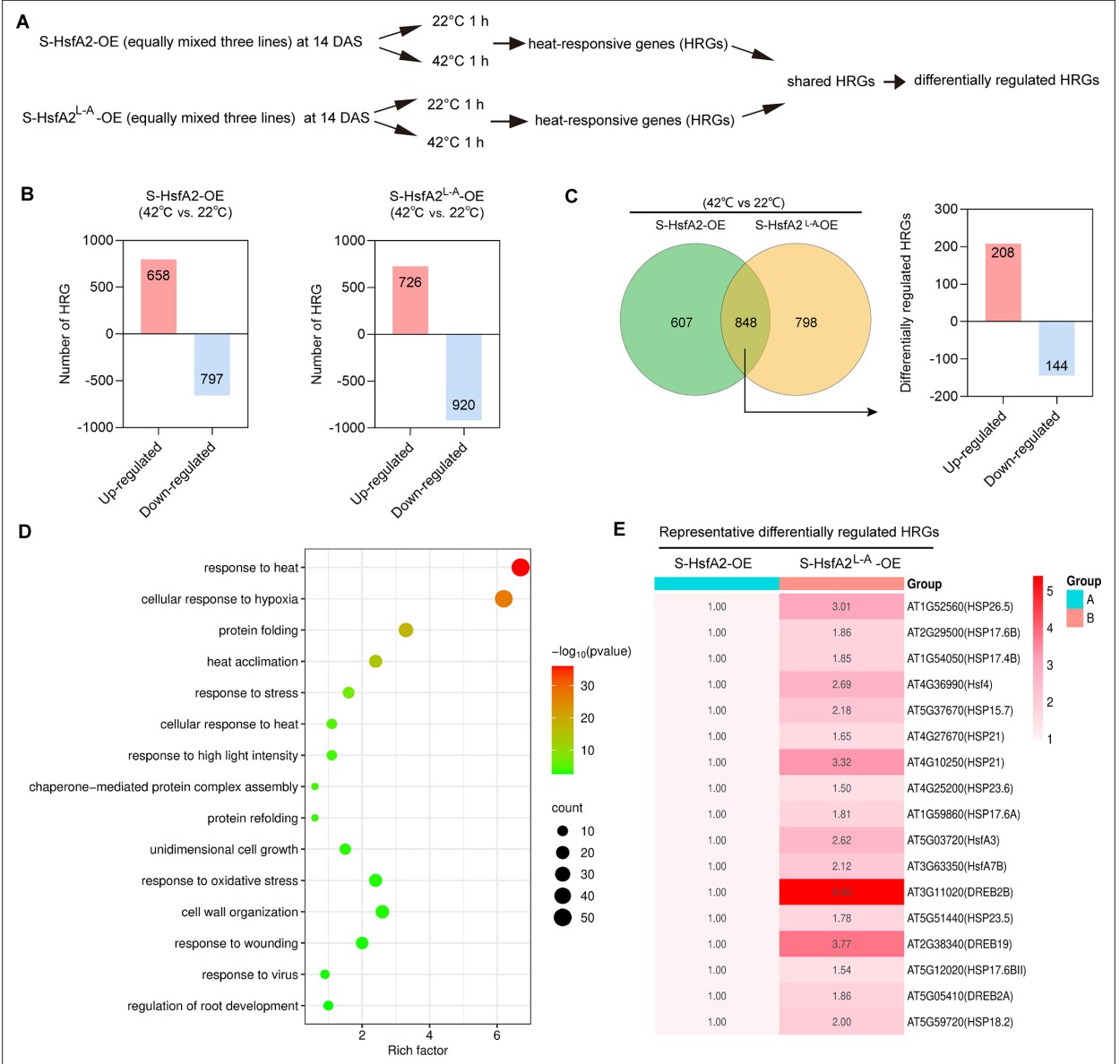

**Figure 2.** Comparative transcriptome analysis between S-HsfA2-OE and S-HsfA2^(L-A)-OE revealed differentially expressed heat-responsive genes. (**A**) Experimental scheme for use in mRNA-Seq analysis. (**B**) The number of heat stress-responsive genes (HRGs) (42/22°C) for S-HsfA2-OE and S-HsfA2^(L-A)-OE. (**C**) Venn diagram showing 848 HRGs shared between S-HsfA2-OE and S-HsfA2^(L-A)-OE, in which differentially expressed HRGs are shown on the left. (**D**) Bubble diagram showing the top 15 pathways associated with the 848 shared HRGs according to the Gene Ontology (GO) enrichment analyses. The *X*-axis represents the enrichment factor (the ratio of the number of genes enriched in the GO pathway or GO term to the number of annotated genes), and the *Y*-axis represents the name of the pathway. The bubble size represents the number of HRGs involved. The bubble colour indicates the enrichment degree (p-value) of the pathway. (**E**) Heatmap showing the expression profiles (S-HsfA2-OE was set as 1) of selected representative differentially expressed HRGs involved in heat tolerance. DAS, days after sowing.

inverted nGAAn blocks sharing a G base (5′-nGAA->G<-AAGn-3′), indicating that the HRE and HSE are partially related. We next confirmed the binding of S-HsfA2 or tDBD to the HRE. The bacterially expressed and purified glutathione S-transferase (GST)-tagged S-HsfA2 fusion protein (GST–S-HsfA2) (*Figure 3—figure supplement 1*) was subjected to electrophoretic mobility shift assays (EMSAs). A strong nonspecific shift band originating from the GST control was observed, which was also present in reactions containing GST–S-HsfA2 with either the HRE probes or the mutated HRE probes. Nevertheless, a weak yet detectable shift band was evident (*Figure 3C*). Moreover, the formation of the GST–S-HsfA2 complex was reduced in the presence of unlabelled HRE competitor DNA (*Figure 3C*), confirming that S-HsfA2 binds specifically to HREs in vitro. According to the yeast one-hybrid (Y1H)

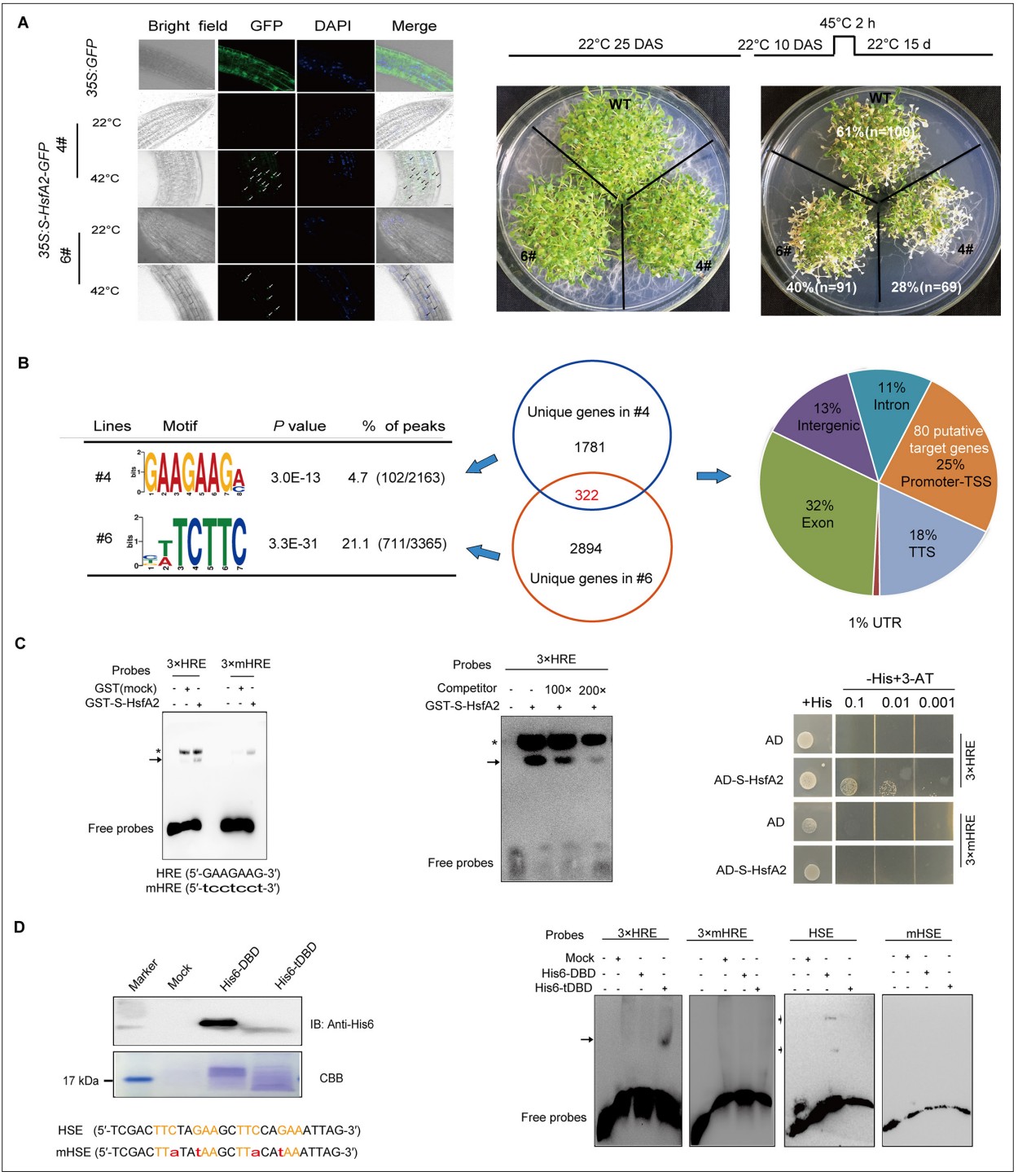

**Figure 3.** S-HsfA2 specifically binds the heat-regulated element (HRE) identified by ChIP-Seq in *35S:S-HsfA2-GFP Arabidopsis* plants. (**A**) Representative images of the subcellular localization of S-HsfA2-GFP in the root cells of two independent transgenic *Arabidopsis* lines (#4 and #6) treated without (normal) or with heat stress (42°C for 1 hr). GFP was used as a negative control. Scale bar, 50 μm. Compared with the wild-type (WT) control, the two lines presented a heat stress-sensitive phenotype (right). (**B**) ChIP-Seq analysis using an anti-GFP antibody in two *35S:S-HsfA2-GFP* lines revealed a centrosymmetric 7 bp motif and 80 putative target genes of S-HsfA2. (**C**) S-HsfA2 binding to HREs was verified by electrophoretic mobility shift assay (EMSA) and yeast one-hybrid (Y1H) assays. The mutated HRE (mHRE) was used as a negative control. Unlabelled HRE is used as a competitor. For the EMSAs, the specific binding signal is marked with an arrow, whereas the asterisk indicates a nonspecific signal. EMSAs were repeated two times, and typical images are shown. (**D**) The production of His6-DBD and His6-tDBD was confirmed by immunoblotting with an anti-His6 antibody (left). Mock proteins (empty vector control) were used as a negative control. Coomassie Brilliant Blue (CBB)-stained proteins are shown as loading controls. The

*Figure 3 continued on next page*

*Figure 3 continued*

binding of His6-DBD and His6-tDBD to heat shock element (HSE) or mutated HSE (mHSE) was verified by EMSA (right). The HRE and HSE sequences are shown, and the corresponding mutations are shown in lowercase. DAS, days after sowing.

The online version of this article includes the following source data and figure supplement(s) for figure 3:

**Source data 1.** PDF file containing original electrophoretic mobility shift assays (EMSAs) and western blot for *Figure 3C, D*, showing the relevant bands boxed and labelled.

**Source data 2.** Original files for electrophoretic mobility shift assays (EMSAs) and western blots displayed in *Figure 3C, D*.

**Figure supplement 1.** Bacterially expressed and purified GST–S-HsfA2.

**Figure supplement 1—source data 1.** PDF file containing original western blots for *Figure 3—figure supplement 1*, showing the relevant bands boxed and labelled.

**Figure supplement 1—source data 2.** Original files for western blot analysis displayed in *Figure 3—figure supplement 1*.

**Figure supplement 2.** His6-HsfA2 fails to bind to the heat-regulated element (HRE) in vitro.

**Figure supplement 2—source data 1.** PDF file containing original western blots and electrophoretic mobility shift assay (EMSA) for *Figure 3—figure supplement 2*, showing the relevant bands boxed and labelled.

**Figure supplement 2—source data 2.** Original files for western blots and electrophoretic mobility shift assay (EMSA) analysis in *Figure 3—figure supplement 2*.

---

assay, S-HsfA2 also specifically recognized HREs (*Figure 3C*). Interestingly, His6-tDBD recognized the HRE, but His6-DBD bound to the HSE in vitro (*Figure 3D*). Consistent with this observation, His-tagged HsfA2 did not bind to HREs in vitro (*Figure 3—figure supplement 2*). These findings indicate that S-HsfA2 uses the tDBD to bind to HREs in vitro.

## HRE serves as a heat response element

To determine the heat induction effect of HRE, we constructed an HRE trimer driving the minimal 35S promoter (–46/+8, 35Sm)-β-glucuronidase (*GUS*) reporter gene (*HRE-35Sm:GUS*) in transgenic *Arabidopsis* (*Figure 4A*). HRE could drive 35Sm to enable clear GUS histochemical staining in seedlings. However, as observed for the vector control (*35Sm:GUS*), mutations in the HRE (*mHRE-35Sm:GUS*) caused a full loss of GUS staining regardless of heat stress. Weak GUS signals were detected mainly in the shoot apical region and the shoot vascular system in the *HRE-35Sm:GUS* seedlings (*Figure 4A*). In a time-course study, GUS signals were increased under heat stress (37°C) for 15–30 min, suggesting that GUS might be transported via the vascular system from the shoot apical region to the shoot to respond to the heat response at the whole-plant level. However, we cannot exclude the possibility that the increase in GUS signals could also result from local protein production.

A quantitative *GUS* expression assay at both the mRNA (*Figure 4B*) and protein (*Figure 4C*) levels revealed that HRE conferred heat responsiveness to 35Sm by two- to fourfold. This heat responsiveness was also heat (37°C)-time dependent (*Figure 4D*). Nuclear extracts derived from *Arabidopsis* under normal (22°C) and heat stress (37 and 42°C) conditions bound the biotin-labelled HRE probes to generate two clear shift bands in vitro (*Figure 4E*), further suggesting that HRE has the biochemical characteristics of a heat response element. Overall, the above molecular, biochemical, and genetic data strongly demonstrated that the HRE serves as a heat response element.

## S-HsfA2 acts as an HRE-binding transcription repressor in *Arabidopsis*

To determine whether S-HSfA2 depends on the HRE to regulate a gene, we introduced the *35S:S-HsfA2-Flag* or *35S:S-HsfA2^{L-A}-Flag* effector into the *HRE-35Sm:GUS* reporter lines by crossing, and we found that GUS protein levels were downregulated compared with those in uncrossed controls (*Figure 4F*). In contrast, the *35S:S-HsfA2^{L-A}-Flag* effector increased GUS levels (*Figure 4F*). These results confirmed that S-HsfA2 is an HRE-binding transcription repressor. This finding also indicated that the LxLxLx motif within the LRD is needed for S-HsfA2 repression activity in plant cells, which is consistent with findings in yeast cells (*Figure 1—figure supplement 1*).

## S-HsfA2, S-HsfA4c, and S-HsfB1 represent new kinds of plant HSFs

Given that S-HsfA2 is an HRE-binding transcriptional repressor and confers extreme heat sensitivity in *Arabidopsis*, we hypothesized that S-HsfA2 represents a new HSF. The new HSFs are not limited

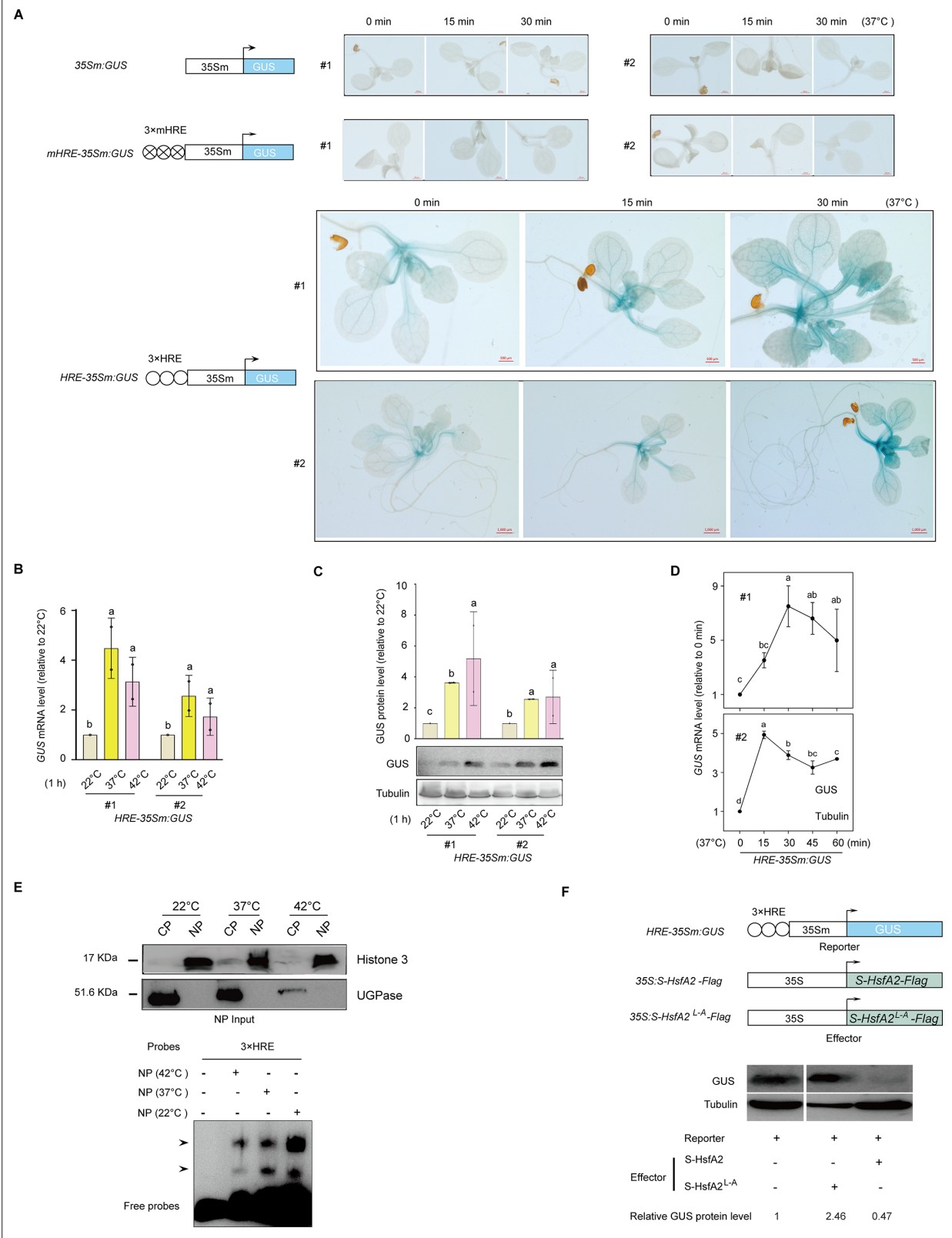

**Figure 4.** Heat-regulated element (HRE) confers heat induction in *Arabidopsis*. (**A**) Qualitative GUS histochemical staining of GUS reporter transgenic *Arabidopsis* seedlings. At least 20 seedlings were analysed, and typical images are shown. Scale bar, 0.5 or 1 mm for *HRE-35Sm:GUS* (#2). Heat-induced GUS activity analyses of two *HRE-35Sm:GUS* transgenic *Arabidopsis* lines at the *GUS* mRNA level via RT−qPCR (**B**) and at the GUS protein level via western blotting with anti-GUS and anti-Tubulin antibodies (**C**). For the *GUS* mRNA level, the control (22°C) was set as 1. The GUS protein level is

*Figure 4 continued on next page*

*Figure 4 continued*

expressed as the relative band intensity of GUS to tubulin (the control (22°C) was set as 1). (**D**) Heat time course analysis of the *GUS* mRNA level in *HRE-35Sm:GUS* transgenic *Arabidopsis* plants. In (**B**) – (**D**), the data are presented as the means ± SDs of at least two independent experiments. Different letters indicate significant differences according to one-way ANOVA, p < 0.05. (**E**) Western blot analysis verified the successful isolation of INPUT nuclear protein (NP) without cytoplasmic protein (CP) contamination via an anti-histone H3 (nuclear marker) antibody and an anti-glucose pyrophosphorylase (UGPase) (cytoplasmic marker) antibody (top). Electrophoretic mobility shift assays (EMSAs) revealed that HRE probes bound to NPs from heat shock-treated (37 and 42°C) or untreated (22°C) *Arabidopsis* plants (bottom). EMSAs were repeated two times, and typical images are shown. The bound complex is indicated by an arrow. (**F**) Abundances of GUS protein in crossed GUS reporter plants harbouring *35S:S-HsfA2-Flag* or *35S:S-HsfA2^{L-A}-Flag* effector (+effector) or an uncrossed reporter control (−effector) were determined by immunoblotting assays using anti-GUS and anti-Tubulin antibodies. The relative GUS expression level in the −effector was set as 1. The data represent the means from two independent experiments.

The online version of this article includes the following source data for figure 4:

**Source data 1.** PDF file containing original electrophoretic mobility shift assay (EMSA) and western blot for *Figure 4C, E, F*.

**Source data 2.** Original files for electrophoretic mobility shift assay (EMSA) and western blot analysis in *Figure 4C, E, F*.

to S-HsfA2 because tDBD is responsible for HRE binding and is highly conserved among S-HSFs. To provide further supporting data, we characterized S-HsfA4c, S-HsfB1, and S-HsfB2a, which were previously identified (*Liu et al., 2013*).

*HsfA4c* (At5G45710) has two splice variants (*Liu et al., 2013*): an intron 2-containing predominant S-*HsfA4c* and a less abundant intron 1-containing full-length *HsfA4c. S-HsfA4c* encodes S-HsfA4c. S-HsfA4c shares similar structural features (LxLxLx motif-containing LRD, *Figure 5A*) with S-HsfA2. *S-HsfA4c* was constitutively expressed, but heat stress increased its expression (*Figure 5A*). S-HsfA4c acts as a transcriptional repressor in yeast cells, and LRD is needed for its transcriptional repression (*Figure 5—figure supplement 1*). S-HsfA4c bound to the HRE via both Y1H (*Figure 5B*) and EMSA (*Figure 5C*). Overexpression of *35S:S-HsfA4c-GFP* (S-HsfA4c-OE) resulted in heat sensitivity compared with that of the *35S:GFP* vector control (*Figure 5D*), whereas T-DNA insert-mediated knockdown of *S-HsfA4c* together with *HsfA4c* increased thermotolerance (*Figure 5E*). Like S-HsfA2-GFP (*Figure 2A*), S-HsfA4c-GFP was detected mainly in the nucleus after heat stress (*Figure 4F*).

Like *S-HsfA4c, S-HsfB1* was constitutively expressed, but heat stress strongly increased its expression (*Figure 5—figure supplement 2*). In contrast, *S-HsfB2a* expression was heat stress inducible (*Figure 5—figure supplement 2*). S-HsfB1 and S-HsfB2a were localized to the nucleus of plant cells (*Figure 5—figure supplement 2*). S-HsfB2a is a weak transcriptional activator in yeast cells, but S-HsfB1 has no transactivation activity (*Figure 5—figure supplement 3*). According to the Y1H results, both S-HsfB1 and S-HsfB2a could bind to the HRE (*Figure 5—figure supplement 2*). However, S-HsfB1, but not S-HsfB2a, bound to the HRE in vitro (*Figure 5—figure supplement 2*). Thus, S-HsfB2a did not appear to be an HRE-binding protein. Therefore, we selected S-HsfB1 for further confirmation of its effects on *Arabidopsis* thermotolerance. We found that seedling survival rate, especially chlorophyll content, was markedly reduced by S-HsfB1 overexpression (*35S:S-HsfB1-RFP*) in *Arabidopsis* plants under heat stress (*Figure 5—figure supplement 2*). In contrast, the specific knockdown of *S-HsfB1* through antisense RNA (targeting the retained intron sequences) and both the *HsfB1* and *S-HsfB1* double knockout resulted in increased seedling survival, especially in terms of the chlorophyll content, under heat stress (*Figure 5—figure supplement 2*).

Collectively, these data indicate that HRE binding, nuclear localization, and negative regulatory effects on extreme heat stress tolerance are common for S-HsfA2, S-HsfA4c, and S-HsfB1. Therefore, we conclude that these three S-HSFs represent new kinds of plant HSFs.

## HRE–HRE-like and HSE elements mediate the *HSP17.6B* promoter's heat response

Using S-HsfA2, we further investigated the transcriptional cascades linking S-HsfA2 to *Arabidopsis* heat tolerance sensitivity. Among the 80 putative targets of S-HsfA2, we focused on the heat shock-induced small *HSP* gene *HSP17.6B* (At2G29500) (*Scarpeci et al., 2008*). This gene is also an upregulated HRG in S-HsfA2^{L-A}-OE vs. S-HsfA2-OE (*Figure 2E*). By searching the ChIP-Seq dataset, we identified the 103-bp promoter region of *HSP17.6B*. This region contains 14-bp DNA sequences (5'-GAAGAAGGAAGAAC-3', −540 to −527 relative to the transcription start site), which consists of one perfect HRE (5'-GAAGAAG-3') and one HRE-like (5'-GAAGAAC-3'), named the HRE–HRE-like element (*Figure 6A*). HSP17.6Bp also contains a perfect HSE (5'-aTTCtaTTCaaTTCa-3', −94 to −80, *Figure 6A*).

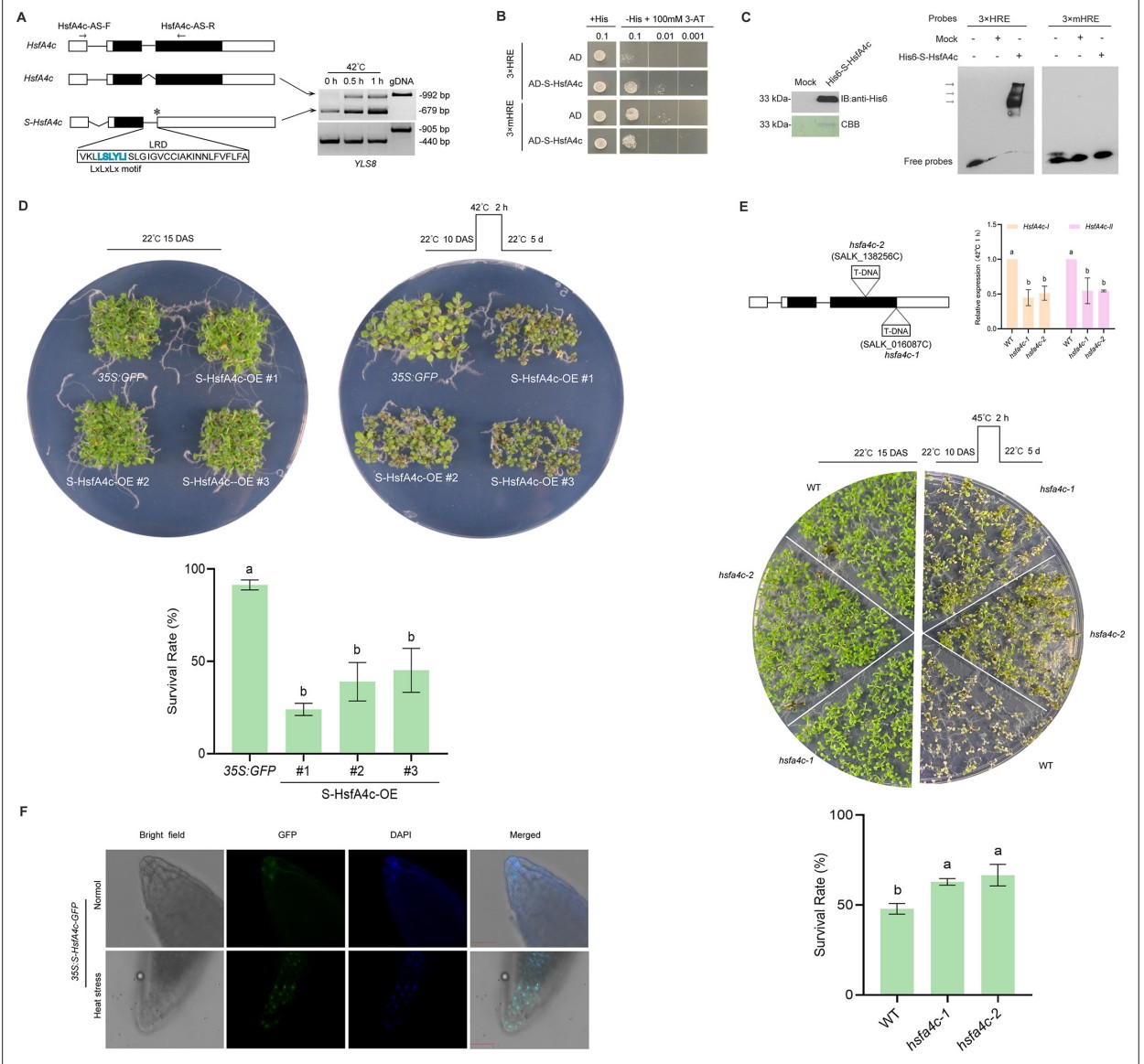

**Figure 5.** S-HsfA4c is similar to S-HsfA2 and represents a new HSF. (**A**) Schematic diagrams of *HsfA4c* splice variants and their RT–PCR splicing analyses in 2-week-old *Arabidopsis* plants under heat stress. The *YLS8* gene served as a loading control. gDNA, genomic DNA control. The asterisk indicates the in-frame stop codon. S-HsfA4c binding to the heat-regulated element (HRE) was verified by yeast one-hybrid (Y1H) (**B**) and electrophoretic mobility shift assay (EMSA) with bacterially expressed and purified His6-S-HsfA4c (**C**). The above experiments were performed two times, each yielding similar results. (**D**) Thermotolerance of the *35S:GFP* vector control and *35S:S-HsfA4c-GFP* overexpression (S-HsfA4c-OE) lines. A representative image is shown (left). The survival rates of the seedlings are presented as the means ± SDs of two independent experiments. Different letters indicate significant differences according to one-way ANOVA, $p < 0.05$. (**E**) Thermotolerance of the wild-type (WT) control and two T-DNA *HsfA4c* insert-knockdown mutants (*hsfa4c-1* and *hsfa4c-2*). Abundances of *HsfA4c-I* and *HsfA4c-II* in *hsfa4c-1* and *hsfa4c-2* were confirmed by RT–qPCR (top), a representative image (middle), and survival rate analysis (bottom). The data are presented as the means ± SDs of three independent experiments. Different letters indicate significant differences according to one-way ANOVA, $p < 0.05$. (**F**) Representative images of the subcellular localization of S-HsfA4c-GFP in *Arabidopsis* root cells treated without (normal) or with heat stress (42°C for 1 hr). DAPI, 4,6-diamidino-2-phenylindole (nuclei staining). Scale bar, 40 µm. DAS, days after sowing.

The online version of this article includes the following source data and figure supplement(s) for figure 5:

**Source data 1.** PDF file containing original electrophoretic mobility shift assay (EMSA) and western blot for *Figure 5C*, showing the relevant bands boxed and labelled.

**Source data 2.** Original files for western blot and electrophoretic mobility shift assay (EMSA) analysis in *Figure 5C*.

**Figure supplement 1.** The LRD is responsible for S-HsfA4c transcriptional repression.

*Figure 5 continued on next page*

*Figure 5 continued*

**Figure supplement 2.** S-HsfB1 is similar to S-HsfA2 and represents a new HSF.

**Figure supplement 2—source data 1.** PDF file containing original western blots and electrophoretic mobility shift assay (EMSA) for *Figure 5—figure supplement 2D, E*, showing the relevant bands boxed and labelled.

**Figure supplement 2—source data 2.** Original files for western blots and electrophoretic mobility shift assay (EMSA) analysis in *Figure 5—figure supplement 2D, E*.

**Figure supplement 3.** Transactivation activity of S-HsfB1 and S-HsfB2a in yeast cells.

**Figure supplement 4.** Overexpression of S-HsfA2-GFP, S-HsfA4c-GFP, or S-HsfB1-RFP inhibits the transgenic *Arabidopsis* root growth.

To determine the role of two elements related to the heat response in the response of HSP17.6Bp to heat stress, we generated transgenic *Arabidopsis* lines harbouring HSP17.6Bp, HSP17.6Bp lacking the HRE–HRE-like element (HSP17.6BpΔHRE), HSP17.6Bp lacking the HSE element (HSP17.6BpΔHSE), or HSP17.6Bp lacking both elements (HSP17.6BpΔHREΔHSE). GUS histochemical staining revealed that these promoters conferred a *GUS* reporter expression response to heat (37°C) stress (*Figure 6A*). Compared with the HSP17.6Bp control, the HRE-HRE-like deletion increased the heat-induced *GUS* mRNA level at 37°C, the HSE deletion decreased it at 42°C, and the deletion of these two elements largely decreased the heat-induced GUS expression level. These results suggest that the HRE-HRE-like and HSE elements are responsible for the response of HSP17.6Bp to heat stress and that the HRE-HRE-like element negatively regulates HSP17.6Bp at 37°C, whereas the HSE is positive at 42°C.

## S-HsfA2 negatively regulates *HSP17.6B* by binding to the HRE–HRE-like element

ChIP–qPCR assays verified the binding of S-HsfA2 to the HRE–HRE-like-containing HSP17.6Bp region (*Figure 6B*). Correspondingly, the heat-induced expression level of *HSP17.6B* was downregulated in the *S-HsfA2*-overexpressing lines but upregulated in the *S-HsfA2*-knockdown lines (*Figure 6C*). These results indicated that S-HsfA2 negatively regulates *HSP17.6B* by directly binding to the HRE–HRE-like element within HSP17.6Bp nuclear localization.

### *HSP17.6B* confers heat tolerance in *Arabidopsis*

We next tested whether *HSP17.6B* is involved in thermotolerance. We obtained a T-DNA-inserted *HSP17.6B* mutant (*hsp17.6b-1*), *35S:HSP17.6B-Flag* transgenic *Arabidopsis* plants in the WT background (HSP17.6B-OE), and *35S:HSP17.6B-Flag* transgenic *Arabidopsis* plants in the *hsp17.6b-1* background (HSP17.6B-KI) (*Figure 6D*). Subsequent heat tolerance phenotype analyses revealed that the *hsp17.6b-1* mutant was heat sensitive, whereas the HSP17.6B-OE lines were heat tolerant (*Figure 6E*). The heat tolerance of the HSP17.6B-KI lines was similar to that of the WT control (*Figure 6E*), suggesting that expressing *35S:HSP17.6B* rescued the heat-sensitive phenotype of the *hsp17.6b-1* mutant.

Overall, *HSP17.6B* confers thermotolerance, but S-HsfA2 represses *HSP17.6B* via the HRE–HRE-like element, which is involved in heat induction by Hsp17.6Bp. Thus, *HSP17.6B* links S-HsfA2 to extreme heat sensitivity in *Arabidopsis*. On the basis of these findings, we propose that a noncanonical HSR, that is, S-HsfA2-HRE-*HSP17.6B*, is involved in extreme heat sensitivity.

### *HSP17.6B* overexpression mediates heat tolerance hyperactivation

We noted that *HSP17.6B* overexpression also retarded seedling growth under normal conditions, as indicated by both decreased biomass (fresh weight) and low chlorophyll content (chlorosis) (*Figure 6F*). Compared with the WT control seedlings, the HSP17.6B-OE seedlings appeared to have a dwarf phenotype when grown in soil (*Figure 6F*). Heat stress resulted in browning of the WT leaves, but the leaves of the HSP17.6B-OE plants remained green (*Figure 6F*), suggesting that the HSP17.6B-OE seedlings also presented a heat-tolerant phenotype in the soil. These results demonstrated that *HSP17.6B* overexpression mediated, to some extent, heat tolerance hyperactivation. Therefore, the noncanonical S-HsfA2-HRE-*HSP17.6B* HSR may attenuate heat tolerance hyperactivation.

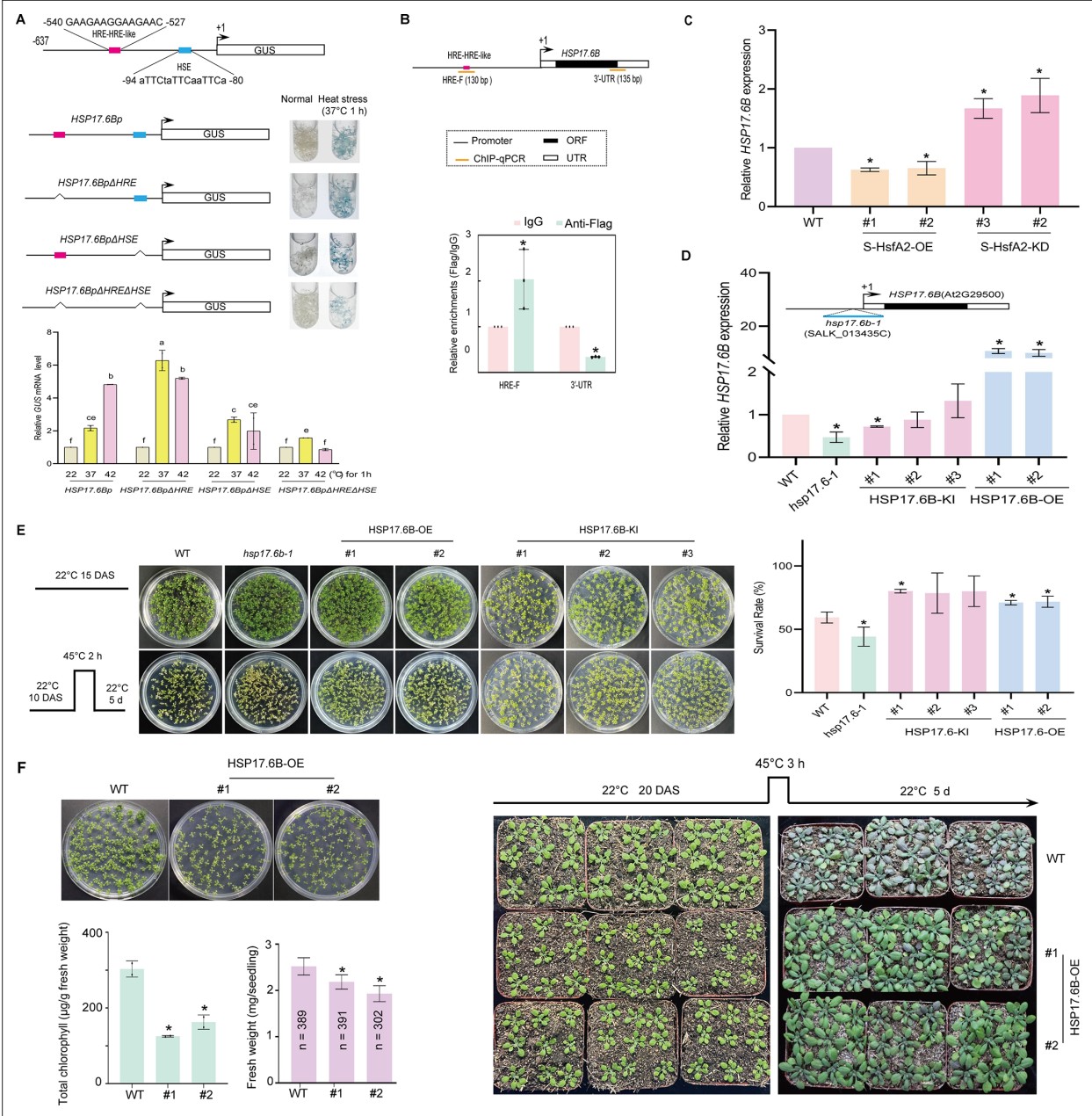

**Figure 6.** Heat-responsive *HSP17.6B* is regulated by S-HsfA2 and confers tolerance in *Arabidopsis*. (**A**) Heat-induced GUS expression assays of *Arabidopsis* plants harbouring a series of HSP17.6Bp-driven *GUS* reporter transgenes via GUS histochemical staining (top) and heat-induced *GUS* mRNA level analysis via RT−qPCR (bottom). For each *GUS* reporter transgene, mixed *Arabidopsis* plants (10 DAS) from three independent transgenic lines were used. The data are presented as the means ± SDs of two independent experiments (the wild-type [WT] control was set as 1). Different letters indicate significant differences according to one-way ANOVA, p < 0.05. (**B**) Schematic diagrams of the *HSP17.6B* gene (top) showing heat-regulated element (HRE)–HRE-like and ChIP−qPCR products. ChIP−qPCR experiments were performed with an anti-Flag antibody and mouse IgG (mock control) in *35S:S-HsfA2-Flag* (S-HsfA2-OE) *Arabidopsis* plants. (**C**) Relative expression levels of *HSP17.6B* (the WT control was set as 1) in S-HsfA2-OE and S-HsfA2-KD plants were analysed via RT−qPCR under heat (45°C for 3 hr) stress. (**D**) Relative expression levels of *HSP17.6B* (the WT control was set as 1) in the T-DNA mutant (*hsp17.6b-1*), three lines of *35S:HSP17.6B-Flag/hsp17.6b-1* (HSP17.6B-KI), and two lines of *35S:HSP17.6B-Flag* (HSP17.6B-OE) were determined via RT−qPCR under heat (37°C for 2 hr) stress. Schematic diagrams of *17.6b-1* are shown (top). (**E**) The survival rates of the WT control, *hsp17.6b-1*, and HSP17.6B-OE and HSP17.6B-KI plants were analysed after heat stress (a representative image is shown on the left). (**F**) HSP17.6B-OE plants (15 DAS) presented defects in growth, as indicated by decreased chlorophyll content and fresh weight (left). The dwarf but heat-resistant phenotype of the HSP17.6B-OE plants grown on soil (right). In (**C**) – (**F**), the data are presented as the means ± SDs of at least two independent experiments. The significance of differences between the experimental values was assessed by Student's *t* test (*p < 0.05). DAS, days after sowing.

The online version of this article includes the following source data and figure supplement(s) for figure 6:

*Figure 6 continued on next page*

*Figure 6 continued*

**Figure supplement 1.** HsfA2 binds to the heat shock element (HSE) of the *HSP17.6B* promoter.

**Figure supplement 1—source data 1.** PDF file containing original western blot for *Figure 6—figure supplement 1A*, showing the relevant bands boxed and labelled.

**Figure supplement 1—source data 2.** Original files for western blot analysis in *Figure 6—figure supplement 1A*.

## Canonical HSR: HsfA2-HSE-*HSP17.6B*

HSP17.6Bp also contains an HSE that is involved in heat induction by HSP17.6Bp (*Figure 6A*). These findings suggest that *HSP17.6B* is regulated by HSFs. The constitutive expression of *HSP17.6B* is activated by *HsfA2* overexpression, but the heat-induced expression of *HSP17.6B* is markedly reduced in the *HsfA2* knockout mutant, suggesting that *HSP17.6B* is a putative target of HsfA2 (*Nishizawa et al., 2006*). ChIP–qPCR of *Arabidopsis* transiently expressing *35S:HsfA2-RFP* with an anti-RFP antibody revealed the direct binding of HsfA2-RFP to the HSE-containing *HSP17.6Bp* region (*Figure 6—figure supplement 1*). According to the Y1H assay results, HsfA2 also bound to the HSE within HSP17.6Bp (*Figure 6—figure supplement 1*). Together with the findings of a previous report (*Nishizawa et al., 2006*), our data confirmed that *HSP17.6B* is a direct target of HsfA2. Thus, a canonical HSR (HsfA2-HSE-*HSP17.6B*) is suggested.

Taken together, the above data indicate that S-HsfA2 and HsfA2 have opposite effects on *HSP17.6B* expression: repression by S-HsfA2 but activation by HsfA2. These findings suggest that the balance between HsfA2 and S-HsfA2 activity serves to fine-tune the *HSP17.6B* expression level, possibly preventing the hyperactivation of heat tolerance mediated by *HSP17.6B* overexpression.

## S-HsfA2 interacts with several HSFs, including HsfA2

S-HSFs can negatively regulate the activities of HSFs through protein–protein interactions (*Wu et al., 2019*; *Zhang et al., 2024*). We therefore explored whether and how S-HsfA2 regulates the activities of HSFs, especially HsfA2, through protein–protein interactions. To address this issue, we first used a yeast two-hybrid (Y2H) assay to screen specific *Arabidopsis* HSFs that interact with S-HsfA2. The results revealed that S-HsfA2 interacted with HsfA1e, HsfA2, HsfA3, HsfA7a, and HsfB2b among the 19 *Arabidopsis* HSFs tested (*Figure 7A*). In this study, we focused on HsfA2. GST pulldown confirmed the interaction between S-HsfA2 and HsfA2 in vitro (*Figure 7B*). Importantly, Y2H and bimolecular fluorescence complementation (BiFC) assays revealed that S-HsfA2 interacted with the DBD of HsfA2 in the nucleus (*Figure 7C*), which is consistent with previous findings that S-ZmHsf17 can interact with the DBD of full-length ZmHsf17 in a Y2H assay (*Zhang et al., 2024*).

## S-HsfA2 decreases the HSE-binding capacity of HsfA2 in vitro

The streptavidin-bead pulldown assay is an efficient in vitro method for evaluating the DNA-binding capacity of transcription factors (*Deng et al., 2003*). Because of the specific interaction between His6-HsfA2 and GST–S-HsfA2 in vitro (*Figure 7B*), we used His6-HsfA2 and GST–S-HsfA2 in this pulldown assay (*Figure 7D*). The results showed that His6-HsfA2 prebinding with GST–S-HsfA2 decreased the signal intensity of the biotin-labelled HSE bait-binding His6-HsfA2 (*Figure 7D*). These findings suggest that S-HsfA2 serves as a negative binding regulator of HsfA2 to decrease the HSE-binding capacity of HsfA2 in vitro.

## S-HsfA2 attenuates HsfA2-regulated *HSP17.6B* promoter expression

We then explored the effects of S-HsfA2, a negative binding regulator of HsfA2, on HsfA2-mediated HSP17.6Bp expression. On the basis of the interaction between HsfA2-nYFP and S-HsfA2-cYFP in the BiFC assay (*Figure 7C*), we used dual-luciferase (LUC) reporter assays to test whether S-HsfA2-cYFP inhibits the *HSP17.6Bp* activity activated by HsfA2-nYFP. To avoid the effect of S-HsfA2 on *HSP17.6Bp*, a short, truncated 135-bp *HSP17.6Bp* (sHSP17.6Bp) containing HSE but lacking an HRE–HRE-like sequence was used in this assay. Compared with the GUS effector controls, HsfA2-nYFP increased *sHSP17.6Bp-LUC* reporter expression in tobacco cells. As expected, S-HsfA2-cYFP failed to regulate sHSP17.6Bp. However, compared with the GUS-cYFP control, S-HsfA2-cYFP reduced the HsfA2-nYFP-mediated *sHSP17.6Bp-LUC* reporter expression level by more than 50% (*Figure 7E*). These results

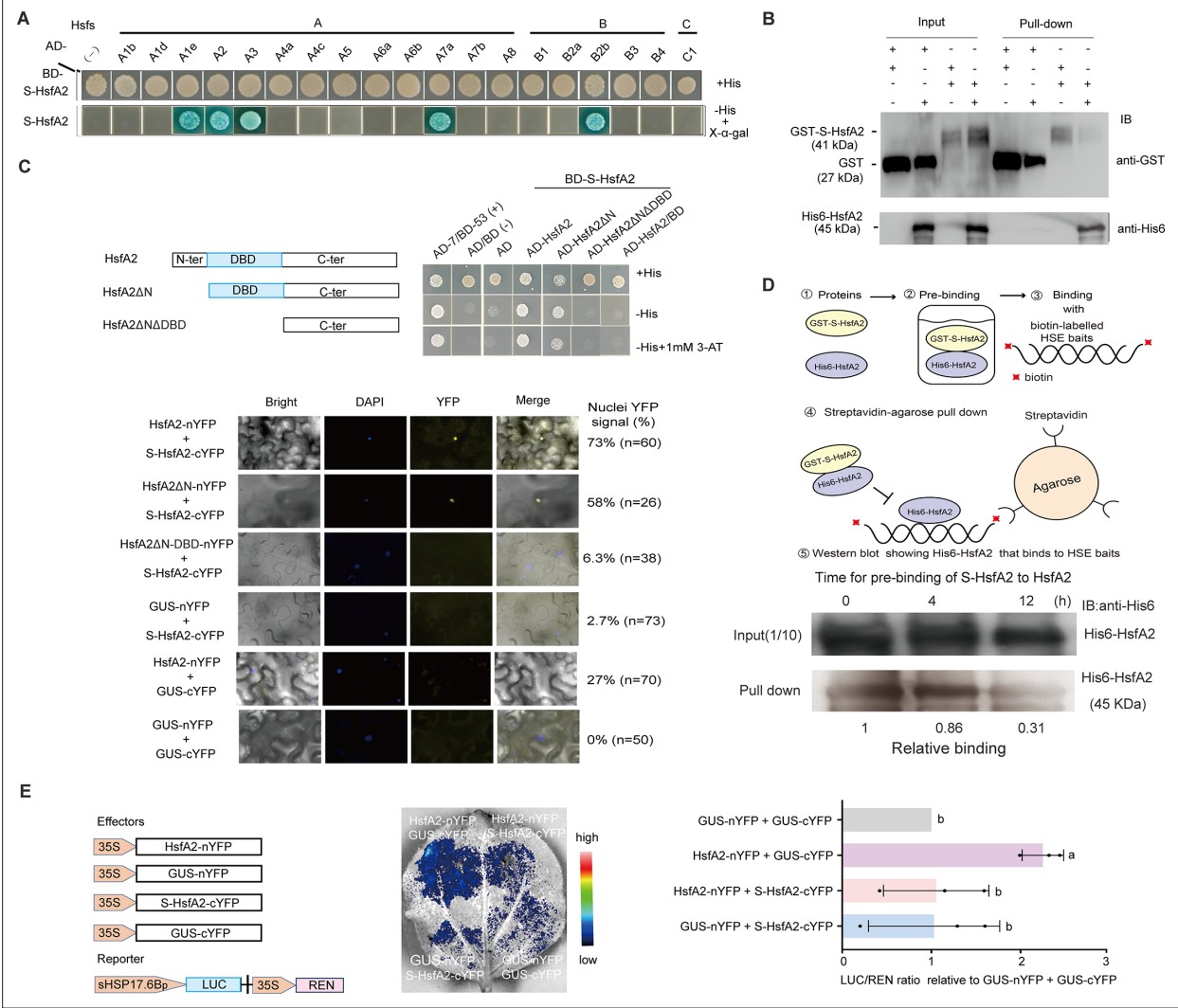

**Figure 7.** fA2 serves as a negative binding regulator of HsfA2 by interacting with the DNA-binding domain (DBD) of HsfA2. (**A**) Five HSFs that interact with S-HsfA2 were identified via yeast two-hybrid (Y2H) analysis of 19 *Arabidopsis* HSFs. (**B**) GST pull-down verified that S-HsfA2 interacted with HsfA2 in vitro. (**C**) Further Y2H and bimolecular fluorescence complementation (BiFC) assays confirmed that S-HsfA2 interacted with the DBD of HsfA2. (**D**) Schematic diagram of the streptavidin–agarose bead pull-down assay used to verify the prebinding of GST–S-HsfA2 with His6-HsfA2 to decrease the biotin-labelled heat shock element (HSE) bait-binding capacity of His6-HsfA2. The data are the means of two independent experiments (the control (0 hr) was set as 1). (**E**) Dual-luciferase (LUC) assays in tobacco leaves. A representative bioluminescence image of the *sHSP17.6Bp:LUC* reporter coexpressing the indicated effector groups. The effector-driven reporter expression activity was expressed as the relative ratio of LUC activity to Renilla luciferase (REN) activity (the GUS effector control was set as 1). The data are presented as the means ± SDs of three independent experiments. Different letters indicate significant differences according to one-way ANOVA, p < 0.05.

The online version of this article includes the following source data for figure 7:

**Source data 1.** PDF file containing original western blot for *Figure 7B, D*, showing the relevant bands boxed and labelled.

**Source data 2.** Original files for western blot analysis in *Figure 7B, D*.

establish the importance of S-HsfA2 in HsfA2-regulated HSP17.6Bp expression by preventing HsfA2 from binding to the HSE.

## Discussion

The AS events of HSFs have been extensively described in plants (*Sugio et al., 2009*; *Liu et al., 2013*; *Cheng et al., 2015*; *Wu et al., 2019*; *Ma et al., 2023*; *Zhang et al., 2024*), but the specific biological functions of these splice variants and their underlying regulatory mechanisms have largely not been

determined. In this study, we demonstrated that *Arabidopsis* S-HsfA2, S-HsfA4c, and S-HsfB1, which are derived from HSF splicing variants, represent new kinds of HSFs. Several molecular and genetic studies support this conclusion: (1) S-HsfA2 and S-HsfA4c are localized in the nucleus, especially after heat stress, reflecting their cellular heat response features; (2) S-HsfA2, S-HsfA4c, and S-HsfB1 have a unique conserved tDBD that binds to the HRE, a new heat response element; and (3) the overexpression of S-HsfA2, S-HsfA4c, or S-HsfB1 confers extreme heat stress sensitivity. We further revealed the molecular mechanisms underlying the ability of S-HsfA2 to prevent canonical HSR hyperactivation caused by *HSP17.6B* overexpression. Therefore, our findings reveal a novel mechanism by which plants balance thermotolerance and growth and offer insight into the breeding of thermotolerant plants.

## Splice variants of HSFs generate new plant HSFs

In this work, we showed that *Arabidopsis* S-HsfA2, S-HsfA4c, and S-HsfB1 are generated from partial or full retention of the conserved intron in the DBD. S-HSFs (such as S-HsfA2, S-HsfA4c, and S-HsfB1) lack all the C-terminal functional domains of HSFs but share a common unique structural feature, that is, tDBD. Another structural feature of S-HSFs is extended motifs or domains encoded by the retained introns. However, their lengths and sequences vary among different S-HSFs and are therefore less conserved than those of the tDBD. For example, lengths of 26 aa, 30 aa, and 13 aa were found for S-HsfA2, S-HsfA4c, and S-HsfB1, respectively. Owing to these differences, S-HsfA2, S-HsfA4c, and S-HsfB1 can be functionally different.

We found that the extended domain, that is, the LRD, allows S-HsfA2 to act as a transcriptional repressor. More importantly, LRD is needed for the functions of S-HsfA2 in heat tolerance regulation. Through comparative transcriptome analysis, we found that several heat tolerance-related genes, including nine *HSPs*, three *HSFs*, and *DREB2A*, are putative HRGs modulated by S-HsfA2. The heat-induced expression levels of these genes were greater in S-HsfA2$^{L-A}$-OE than in S-HsfA2-OE. While HSPs and HSFs are established thermotolerance factors, DREB2A reportedly confers thermotolerance in *Arabidopsis* through directly activating *HsfA3* expression (*Schramm et al., 2008*; *Yoshida et al., 2008*). These findings may further explain why L-to-A mutations in the LxLxLx motif can rescue the negative role of S-HsfA2 in the tolerance of *Arabidopsis* to heat stress. Therefore, we concluded that the extended motifs or domains are essential for the functions of S-HSFs.

Although S-HsfA2, S-HsfA4c, and S-HsfB1 have the conserved tDBD sequences plus adjacent extended motifs or domains, these small HSF isoforms are of biological importance since they negatively regulate extreme heat tolerance in *Arabidopsis*. Notably, two aspects related to heat stress are consistent with their biological relevance. First, *S-HsfA2, S-HsfA4c*, and *S-HsfB1* strongly respond to extreme heat stress, indicating that they function in extreme heat stress in *Arabidopsis*. We also found that the constitutive expression of S-HsfA2 inhibits *Arabidopsis* growth. Therefore, *Arabidopsis* plants do not produce S-HsfA2 under normal conditions to avoid growth inhibition. Similar to the constitutive expression of S-HsfA2-GFP, the constitutive expression of S-HsfA4c-GFP or S-HsfB1-RFP resulted in extreme heat stress sensitivity in *Arabidopsis* but inhibited root growth (*Figure 5—figure supplement 4*). It is of interest to explore whether S-HsfA4c and S-HsfB1 participate in the same biological process, such as heat tolerance and growth, through the coregulation of downstream genes.

In addition, S-HsfA2-GFP and S-HsfA4c-GFP were more easily detected under heat stress than under nonstress. These findings suggest that S-HsfA2 or S-HsfA4c is possibly regulated post-transcriptionally and/or post-translationally. A detailed analysis of the underlying mechanism involved in S-HsfA2 or S-HsfA4c stability is therefore still warranted, thus allowing us to characterize how S-HSFs respond to heat stress at the protein level. Overall, our data strongly support that S-HSFs act as new kinds of HSFs and thus increase the diversity of HSFs.

Surprisingly, AS events in the DBD of plant HSFs have not been detected in animal HSFs. Several reports have shown that animal HSFs utilize exon skipping to generate HSF splicing isoforms containing full-length DBDs (*Tanabe et al., 1999*; *Fujikake et al., 2005*; *Zhang et al., 2001*; *Neueder et al., 2014*). These animal HSF isoforms have different transcriptional activities and work synergistically to regulate the transcription of *HSPs* (*Tanabe et al., 1999*; *Neueder et al., 2014*). Therefore, new kinds of HSFs seem to be plant specific, reflecting a unique feature of plant systems.

## HRE is a heat response element

HSE is a known heat-responsive element and is essential for heat-inducible transcription of genes. In this study, we showed that the HRE is a heat response element, as determined by molecular, biochemical, and genetic analyses. HRE confers a minimal promoter response to heat and is responsible for heat-induced expression of the *HSP17.6B* promoter. *He et al., 2022* reported a heat stress sensing and transmission pathway at the whole-plant level. Nitric oxide (NO) bursts in the shoot apex under heat stress, and NO, an *S*-nitrosoglutathione, can rapidly move from shoot to root through the vascular system in whole *Arabidopsis* plants. Interestingly, the GUS reporter driven by HREs was constitutively expressed in the shoot apical region, and heat stress increased and promoted GUS movement along the shoot vascular system to the leaves and hypocotyl. Therefore, the HRE exhibits heat stress sensing and transmission patterns, indicating a heat regulation function.

The HRE consists of two inverted HSE blocks (nGAAn) that share a G base. Although HRE is partially related to HSE, it does not bind to the DBD or to HsfA2 in vitro. In contrast, HRE is recognized by tDBD, S-HsfA2, S-HsfA4c, and S-HsfB1 in vitro. Loss of the wing domain within the DBD might partly explain why the DNA-binding characteristics of the tDBD change.

We also noted that HREs can drive the constitutive expression of the *GUS* reporter and bind nuclear extracts under normal conditions. These data suggest that some unknown constitutive transcription factors other than S-HSFs exist in *Arabidopsis*. In the future, these HRE-binding transcription factors need to be screened by Y1H and DNA–protein pulldown assays. These studies can further characterize the working molecular mechanisms of HREs.

## The noncanonical HSR: S-HsfA2-HRE-*HSP17.6B*

We propose the S-HsfA2-HRE-*HSP17.6B* noncanonical HSR. In this HSR, S-HsfA2 binds to the HRE–HRE-like element to prevent *HSP17.6B* overexpression and eventually attenuates *Arabidopsis* tolerance to extreme heat. Given that *HSP17.6B* overexpression-mediated heat tolerance hyperactivation represses *Arabidopsis* growth, our results underscore the biological significance of this noncanonical HSR: preventing plant heat tolerance hyperactivation to maintain proper growth. Although we propose a noncanonical HSR based on S-HsfA2, S-HSF-mediated noncanonical HSRs should constitute a widespread regulatory mechanism in planta because the tDBD is highly conserved in S-HSFs and because the HRE can be bound by the tDBD.

## S-HSFs act as new negative binding regulators of HSFs

During the canonical HSR, cells often utilize negative binding regulators of HSFs, including HSP70 and HSF-binding proteins (HSBPs), to inactivate HSFs in different ways (*Morimoto, 1998*). HSP70 represses the transcriptional activity of HSFs by directly binding to the transcriptional activation domain (*Baler et al., 1996*). HSBPs are conserved small nuclear proteins that dissociate trimeric HSFs and abate transcription activation (*Satyal et al., 1998*). *Arabidopsis* HSBP (At4G15802) attenuates the canonical HSR by interacting with HSFs and decreasing HSF DNA-binding activity (*Hsu et al., 2010*). S-ZmHsf17 can interact with the DBD of ZmHsf17 to suppress the transactivation of ZmHsf17 by reducing the HSE-binding capacity of ZmHsf17 (*Zhang et al., 2024*), suggesting that S-HSFs may act as negative binding regulators of HSFs. In the present study, S-HsfA2 was also identified as a negative binding regulator of HsfA2. Unlike HSP70 and HSBPs, S-HsfA2 binds to the DBD, thus decreasing the HSE-binding capacity of HSFs such as HsfA2 and eventually attenuating HsfA2-regulated *HSP17.6B* promoter activity.

## S-HsfA2 molecular working model

Taken together, our findings in this study support the use of a molecular model in which S-HsfA2 balances the gain and loss of canonical HsfA2-regulated *HSP17.6B* overexpression-mediated HSR hyperactivation. S-HsfA2 directly binds to the HRE of the *HSP17.6B* promoter to repress *HSP17.6B* expression and interacts with the DBD of HsfA2 to inactivate HsfA2 binding to the *HSP17.6B* promoter, ultimately alleviating hyperactivation of the HsfA2-HSE-*HSP17.6B* HSR (*Figure 8*). Considering that S-HsfA2 generation occurs upon exposure to severe heat, this negative regulatory pathway is not an 'enemy' but rather a 'friend' to avoid canonical HSR hyperactivation in *Arabidopsis*.

However, the mechanical mechanism by which S-HsfA2 regulates heat tolerance and growth balance may not be limited to *HSP17.6B*. Among the putative target genes, with the exception of

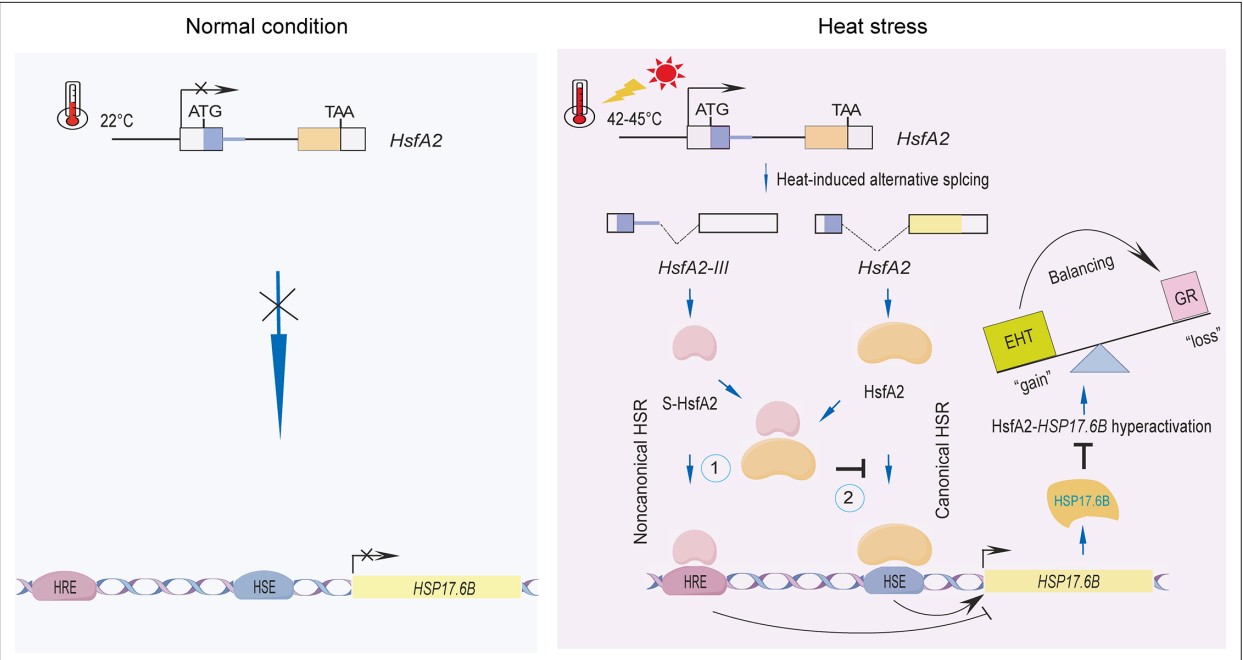

**Figure 8.** A proposed model for S-HsfA2 balancing *Arabidopsis* heat tolerance and growth. Under normal conditions, *HsfA2* and its target *HSP17.6B* gene are not expressed. However, these two genes are induced by heat stress. Furthermore, *HsfA2* undergoes alternative splicing under severe heat stress to generate the splicing transcript *HsfA2-III*, which is translated to S-HsfA2. The overexpression of *HSP17.6B* not only confers thermotolerance but also retards seedling growth, reflecting the hyperactivation of the heat stress response (HSR) mediated by *HSP17.6B* overexpression. S-HsfA2 balances the gain (extreme heat tolerance, EHT) and loss (growth retardation, GR) of HsfA2-regulated *HSP17.6B* overexpression to ensure proper growth in two ways: ① antagonistic repression of *HSP17.6B* overexpression through a noncanonical HSR and ② a negative binding regulator of HSFs that inactivates the *HSP17.6B* promoter-binding capacity of HsfA2.

*HSP17.6B*, the other 79 genes have not been reported to be involved in heat tolerance. We noted that S-HsfA2 can interact with five HSFs, including HsfA3, in Y2H assays. It has been reported that the overexpression of *HsfA3* increases heat tolerance but inhibits *Arabidopsis* growth under nonstress conditions and that *HsfA3* is a downstream target gene of DERB2A (*Yoshida et al., 2008*). We found that the heat-induced expression levels of *DERB2A* and *HsfA3* were greater in S-HsfA2$^{L-A}$-OE than in S-HsfA2-OE, indicating that *DERB2A* and *HsfA3* can be regulated by the transcriptional activity status of S-HsfA2. Therefore, the putative S-HsfA2-*DERB2A*-*HsfA3* module might be associated with the roles of S-HsfA2 in heat tolerance and growth balance. These working mechanisms warrant further investigation.

## Our findings could lead to the breeding of thermotolerant plants

It has been reported that natural variation in *HsfA2* pre-mRNA splicing is associated with changes in thermotolerance during tomato domestication (*Hu et al., 2020*). In the present study, selective knockdown of S-HsfA2 improved tolerance to transient extreme heat (45°C for 2 hr). Given that S-HsfA2 favours *Arabidopsis* growth under extreme temperature conditions by inhibiting heat tolerance hyperactivation mediated by *HSP17.6B* overexpression, our findings offer insights for plant breeding to orchestrate plant extreme heat (such as heat waves) tolerance and growth via extreme heat-specific expression of *S-HsfA2* homologues.

## Materials and methods
### Plant materials and growth conditions
The *hsfa4c-1* mutant (SALK_016087C), the *hsfa4c-2* mutant (SALK_138256C), *hsp17.6b-1* (SALK_013435C), and *hsfb1* (SALK_104713C) were purchased from AraShare (Fuzhou, China) and verified by PCR and RT−qPCR. The seeds of *Arabidopsis* (*A. thaliana*, Columbia-0) and the mutants were surface sterilized with sodium hypochlorite (25%, vol/vol) and washed with sterile double

distilled water. After vernalization treatment for 3 days, the seeds were grown on half Murashige and Skoog (MS) agar plate media with a 16/8 hr light/dark cycle at 22°C and a white light intensity of 150 mmol m$^{-2}$ s$^{-1}$. In this study, the age of *Arabidopsis* seedlings was calculated as days after sowing (DAS).

## Primers

The primers and synthesized oligonucleotides used in this study are listed in **Supplementary file 6**.

## Recombinant plasmid construction

The target DNA fragments were obtained via PCR with the Taq enzyme (Takara, Japan) and subsequently cloned and inserted into the corresponding vectors with the ClonExpress II One Step Cloning Kit (Vazyme Biotech, Nanjing, China).

The S-HsfA2 and S-HsfA2$^{L-A}$ coding sequences lacking stop codons were amplified from *Arabidopsis* genomic DNA via S-HsfA2-Flag-F/-R and S-HsfA2-Flag-F/S-HsfA2mut-Flag-R, respectively. Similarly, the coding sequence of *HSP17.6B* lacking the stop codon 'TGA' was amplified via HSP17.6B-Flag-F/-R. The PCR products were cloned and inserted into the *Xba*I/*Bam*HI site *35S:3×Flag* (our laboratory) and sequenced via GUS-R to generate *35S:S-HsfA2-Flag* (S-HsfA2-OE), *35S:S-HsfA2$^{L-A}$-Flag* (S-HsfA2$^{L-A}$-OE), and *35S:HSP17.6B-Flag* (HSP17.6B-OE).

We used a fusion PCR method to generate the *35S:S-HsfA2-RNAi* construct via pUCCRNAi. First, the PCR1 products (forward intron 1a of *HsfA2*, 104 bp) were amplified from the BD-S-HsfA2 plasmid via the primers F1/R1. Second, the PCR2 products (intron 1 of potato GA20 oxidase, 219 bp) were amplified from pUCCRNAi via the primers F2/R2. Third, the PCR3 products (reverse intron 1a of *HsfA2*, 105 bp) were amplified from the BD-S-HsfA2 plasmid via the primers F3/R3. Fourth, fusion PCR was performed in a total volume of 50 μl containing 33 ng of each purified PCR product (template) for 5 cycles at an annealing temperature of 55°C (without primers) followed by 25 cycles (adding F1/R3) at 65°C. The fusion fragments were cloned and inserted into the *Bam*HI/*Sac*I site of pBI121 and sequenced via GUS-R.

For the expression vector constructs, HsfA2-RFP-F/-R, S-HsfB1-RFP-F/-R, and S-HsfB2a-RFP-F/-R were used to amplify the coding regions of HsfA2, S-HsfB1, and S-HsfB2a from cDNA, which were subsequently cloned and inserted into the *Bam*HI/*Spe*I sites of pCAMBIA1300. The resulting *35S:HsfA2-RFP*, *35S:S-HsfB1-RFP*, and *35S:S-HsfB2a-RFP* were sequenced via RFP-R. The expression vectors *35S:S-HsfB1* and *35S:S-HsfB2a* were constructed via antisense RNA knockdown technology and reverse primers anti-S-HsfB1-F/-R and anti-S-HsfB2a-F/-R. The coding region of S-HsfA4c lacking the stop codon TAG was amplified from cDNA via the primers S-HsfA4c-GFP-F/-R and cloned and inserted into the *Bgl*II/*Spe*I sites of pCAMBIA1302 (Clontech) to generate *35S:S-HsfA4c-GFP*. The coding region of S-HsfA2 lacking the stop codon TAG was amplified from genomic DNA via the primers S-HsfA2-GFP-F/-R and cloned and inserted into the *Nco*I/*Spe*I sites of pCAMBIA1302 (Clontech) to generate *35S:S-HsfA2-GFP*.

For the AD constructs, the coding sequences of 19 HSFs, S-HsfA2, and S-HsfA4c were amplified from heat-treated *Arabidopsis* cDNA via the corresponding primers, inserted into the *Bam*HI/*Bgl*II site of pGAD424 (Clontech) to generate corresponding AD-HSF fusion vectors and identified by sequencing.

For bait construction, 3×HRE, 3×mHRE, HSE (**Enoki and Sakurai, 2011**), and mHSE were synthesized by Shanghai Sangon Biotechnology and subsequently cloned and inserted into pHIS2.1 (Clontech). The PCR products were cloned and inserted into the *Eco*RI/*Spe*I site of the pHIS2.1 vector (Clontech) and sequenced via pHIS2.1 forward or reverse primers.

For the BD constructs, the S-HsfA2 and S-HsfA2$^{L-A}$ coding sequences were amplified from *Arabidopsis* genomic DNA via BD-S-HsfA2-F/-R and BD-S-HsfA2-F/BD-S-HsfA2mut-R, respectively. The S-HsfA4c, S-HsfA4cΔLRD, S-HsfB1, and S-HsfB2a coding sequences were amplified from cDNA via BD-S-HsfA4c-F/-R, BD-S-HsfA4c-F/BD-S-HsfA4cΔLRD-R, BD-S-HsfB1-F/-R, and BD-S-HsfB2a-F/-R, respectively. The corresponding PCR products were subsequently cloned and inserted into the *Eco*RI/*Pst*I sites of pGBKT7 (Clontech) to generate the BD-S-HsfA2, BD-S-HsfA2$^{L-A}$, BD-S-HsfA4c, BD-S-HsfA4cΔLRD, BD-S-HsfB1, and BD-S-HsfB2a constructs. T7 primers were used to sequence the BD fusion vectors.

To generate the glutathione *S*-transferase (GST)-tagged expression plasmid pGEX4T-S-HsfA2. Using the BD-S-HsfA2 plasmid as a template, the PCR product was amplified with GST–S-HsfA2-F/-R and subsequently cloned and inserted into the *Bam*HI/*Sal*I sites of pGEX-4T.

For the His6-tagged expression plasmids, the HsfA2, S-HsfA2, S-HsfA4c, S-HsfB1, and S-HsfB2a coding sequences were amplified from cDNA via His6-HsfA2-F/-R, His6-HsfA2-F/His6-S-HsfA2-R, His6-S-HsfA4c-F/-R, His6-S-HsfB1-F/-R, and His6-S-HsfB2a-F/-R, respectively. The PCR products were inserted into the *Bam*HI/*Sal*I sites of pET-28a(+) to generate His6-HsfA2, His6-S-HsfA2, His6-S-HsfA4c, His6-S-HsfB1 and His6-S-HsfB2a. Then, using His6-HsfA2 and His6-S-HsfA2 as templates, the DBD and tDBD coding sequences were amplified with His6-DBD-F/-R and His6-DBD-F/His6-tDBD-R. The PCR products were also cloned and inserted into the *Bam*HI/*Sal*I sites of pET-28a(+) to obtain His6-DBD and His6-tD. The plasmids were subsequently sequenced via the T7 primer.

We used an asymmetric overlap extension PCR method to construct the *HRE-35Sm:GUS* and *mHRE-35Sm:GUS* plasmids via 3×HRE-35Sm-F or 3×mHRE-35Sm-F and the common reverse primer HRE-35Sm-R. The PCR products were subsequently cloned and inserted into the *Hin*dIII/*Xba*I sites of pBI121 instead of the 35S promoter. The recombinant reporter plasmids were subsequently sequenced via GUS-R primers for validation.

The WT *HSP17.6B* promoter (HSP17.6Bp, –637/+1) was amplified from *Arabidopsis* genomic DNA via HSP17Bp-F/-R. The resulting PCR product was subsequently cloned and inserted into the *Hin*dIII/*Bam*HI sites of pBI121 and subsequently sequenced via GUS-R to generate *HSP17.6Bp:GUS*. On the basis of the *HSP17.6Bp:GUS*, fusion PCR was performed to generate *HSP17.6BpΔHRE:GUS, HSP17.6BpΔHSE:GUS,* and *HSP17.6BpΔHREΔHSE:GUS* via HSP17BpΔHRE-F/-R, HSP17.6BpΔHSE-F/-R, and HSP17.6BpΔHREΔHSE-R, respectively. The above PCR products were cloned and inserted into the *Hin*dIII/*Bam*HI sites of pBI121 and sequenced by GUS-R.

A short, truncated 135-bp *HSP17.6B* promoter fragment (sHSP17.6Bp) was amplified from *Arabidopsis* genomic DNA via sHSP17.6Bp-LUC-F/-R and inserted into the *Hin*dIII/*Spe*I sites of pGreenII0800-LUC (LUC vectors containing the REN gene under the control of the 35S promoter as an internal control) to generate the reporter vector *sHSP17.6Bp:LUC*.

For the BiFC constructs, the coding regions of S-HsfA2 and HsfA2 and a series of HsfA2 genes progressively truncated from the N-terminus were amplified from AD-S-HsfA2 and AD-HsfA2 via cYFP-S-HsfA2-F/-R, nYFP-HsfA2-F/-R, nYFP-HsfA2ΔN-F and nYFP-HsfA2ΔNΔDBD-F, respectively. The above PCR products were cloned and inserted into the *Xba*I/*Bam*HI sites of modified pCAMBIA1300 containing the cYFP or nYFP coding sequence to generate the corresponding constructs S-HsfA2-cYFP, HsfA2-nYFP, HsfA2ΔN-nYFP, and HsfA2ΔN-DBD-nYFP. The above vectors were identified by sequencing nYFP-R or cYFP-R.

## Plant transformation and crossing

A series of plasmids related to S-HsfA2 (*35S:S-HsfA2-Flag*, *35S:S-HsfA2^{L-A}-Flag*, and *35S:S-HsfA2-RNAi*), GUS reporter constructs *HRE-35Sm:GUS*, *mHRE-35Sm:GUS,* and *35Sm:GUS* (**Wang et al., 2023**), recombinant vectors for HSP17.6B (*35S:HSP17.6B-Flag*, *HSP17.6Bp:GUS*, *HSP17BpΔHRE:GUS*, *HSP17BpΔHSE:GUS*, and *HSP17BpΔHREΔHSE:GUS*), and GFP fusion vectors (*35S:S-HsfA2-GFP* and *35S:S-HsfA4c-GFP*) were introduced into the *Agrobacterium* GV3101 strain, which was subsequently used to transform the Columbia wild type via the flower infiltration method. The transgenic plants used in this study were $T_3$ homozygous plants. The transgenic plants used in this study were $T_3$ homozygous plants.

*35S:HsfA2-RFP* was subsequently introduced into *A. tumefaciens* GV3101. Ten-day-old *Arabidopsis* plants were vacuum infiltrated with *A. tumefaciens* as described by **Marion et al., 2008** and subsequently incubated for 3 days.

The effector lines (♂,*35S:S-HsfA2-Flag*, *35S:S-HsfA2^{L-A}-Flag*) were crossed with GUS reporter lines (♀, *HRE-35Sm:GUS*) to generate $T_1$ seeds via artificial pollination. To select the positive crossing lines, the leaves of T1 seedlings of the crossing lines (+effector) and control lines (−effector) were subjected to PCR analysis via a genomic DNA template. The following primer was used: S-HsfA2-F/121-R. Leaves from 10 independent positive-crossing lines were used for protein isolation. *GUS* expression was detected via western blotting as described below.

## Transcriptional activity assay in yeast cells

The transcriptional activation activity was determined in synthetic defined medium lacking Trp but containing 40 µg/ml 5-bromo-4-chloro-3-indolyl-α-D-galactopyranoside (X-α-gal; Sigma-Aldrich) at 30°C for 3 days. BD and BD-PVERF15 were used as negative and positive controls, respectively. β-Galactosidase activity in liquid cultures was determined via an *o*-nitrophenyl-β-D-galactopyranoside (ONPG) assay in three independent clones.

## Heat stress treatments

In a thermotolerance test, *Arabidopsis* plants (10 or 12 DAS) were incubated at 45°C for 2 hr following recovery at 22°C for 5 or 15 days. In the soil experiments, plants (12 DAS) were planted in soil for 8 days, treated in an incubator at 45°C for 3 hr and returned to 22°C for 5 days.

To prepare nuclear extracts, 14-day-old *Arabidopsis* plants were transferred to filter paper with 1/4 MS liquid media and subjected to heat shock in a temperature incubator (22, 37, and 42°C) for 1 hr.

To analyse GUS mRNA or protein expression levels, 14-day-old *HRE-35Sm:GUS*, m*HRE-35Sm:GUS*, or *35Sm:GUS* transgenic *Arabidopsis* plants were subsequently transferred to filter paper in 1/4 MS liquid media and subjected to heat shock in a temperature incubator (22, 37, and 42°C) for 1 hr. For heat time-course analysis of the *HRE-35Sm:GUS* transgenic *Arabidopsis* plants, samples were taken at various treatment intervals (0, 15, 30, 45, and 60 min).

## Heat tolerance assays

For the survival rate, the true leaves of the plants were chlorotic and considered dead. For the root length assay, the seeds of WT and transgenic *Arabidopsis* plants (i.e., S-HsfA2-OE, S-HsfA2$^{L-A}$-OE, and S-HsfA2-KD) were sown on vertical one-half-length MS agar plates at 22°C for 12 days and then subjected to root length analysis with the ImageJ analysis tool. The detached shoots were subjected to chlorophyll content analysis. The shoot chlorophyll contents were determined at 663 and 645 nm as described previously (*Chen et al., 2024*).

## RNA isolation, cDNA synthesis, and PCR analysis

Total RNA was extracted from plant materials via an RNAprep Pure Plant Kit with on-column DNase digestion (Tiangen Biotech, China) according to the manufacturer's protocol. First-strand cDNA was synthesized from RNA (approximately 2 µg) with oligo(dT) primers according to the instructions of the PrimeScript First Strand cDNA Synthesis Kit (Takara).

RT−PCR splicing analysis was carried out with Ex-Taq polymerase (Takara, Japan) for a total of 25−28 cycles. *YLS8* (AT5G08290) was used as a loading control.

RT−qPCR was performed on an IQ5 Multicolor Real-Time PCR Detection System (Bio-Rad) with the SYBR Premix Ex-Taq Kit (Takara). *EF-1α* (AT1G18070) was used as an internal control. The relative expression levels were analysed via the delta−delta cycle threshold method according to CFX Manager software. A p-value <0.05 indicated a statistically significant difference.

## Transcriptome sequencing

Seeds of the S-HsfA2-OE and S-HsfA2$^{L-A}$-OE plants were sown on 1/2 MS agar plates for 2 weeks (14 DAS). To minimize biological variation, seedlings from three independent transgenic lines of S-HsfA2-OE or S-HsfA2$^{L-A}$-OE were pooled equally before being transferred to ½ MS liquid media. The plants were then subjected to thermal treatments at 22°C (control) or 42°C (heat stress) for 1 hr. Biological duplicates ($n \geq 2$) were collected for each experimental condition.

Total RNA was extracted via a Total RNA Extraction Kit (TRIzol) (B511311, Sangon, China) according to the manufacturer's protocol. The high-quality RNA samples were subsequently submitted to Sangon Biotech (Shanghai, China) for library preparation and sequencing. The library fragments were purified with the AMPure XP system (Beckman Coulter, Beverly, USA). Paired-end sequencing of the libraries was performed via NovaSeq sequencers at Sangon Biotech (Shanghai, China). All the raw RNA sequencing data have been deposited in the NCBI BioProject accession (PRJNA1268688).

The raw paired-end reads were trimmed and quality controlled by SeqPrep (https://github.com/jstjohn/SeqPrep; *St John, 2011*) and Sickle (https://github.com/najoshi/sickle; *Joshi and Fass, 2015*) with default parameters. The clean reads were subsequently independently aligned to the reference

genome in orientation mode via HISAT2 (http://ccb.jhu.edu/software/hisat2/index.shtml; *Park et al., 2020*) software. The mapped reads of each sample were assembled via StringTie (https://ccb.jhu.edu/software/stringtie/index.shtml?t=example; *Pertea, 2025*).

Bioinformatics analysis was performed on the clean reads obtained to calculate the expression level of each transcript on the basis of the transcripts per million reads (TPM). Genes exhibiting |Fold-Change|≥ 1 and *q* values <0.05 were identified as heat-responsive genes (HRGs) via DEGseq2 for S-HsfA2-OE or S-HsfA2$^{L-A}$-OE. For comparative analysis between S-HsfA2$^{L-A}$-OE and S-HsfA2-OE, |log$_2$(fold change)| ≥ 0.5 was applied to define differentially regulated HRGs. GO enrichment analysis was performed via DAVID (https://davidbioinformatics.nih.gov/) bioinformatics tools, with q values ≤0.05 considered statistically significant.

## Subcellular localization

For GFP, S-HsfA2-GFP, and S-HsfA4c-GFP localization in *Arabidopsis* cells, the roots of 7-day-old *35S:S-HsfA2-GFP* and *35S:S-HsfA4c-GFP* transgenic *Arabidopsis* plants were observed via fluorescence microscopy and laser confocal microscopy (SP8) in the GFP channel. The cell nuclei were stained with 4,6-diamidino-2-phenylindole (DAPI, C0060; Solarbio). The fusion protein expression vectors *35S:S-HsfB1-RFP* and *35S:S-HsfB2a-RFP* were transiently transformed into tobacco. After 48 hr, the leaf cells were stained with the nuclear stain DAPI under a fluorescence microscope in the RFP channel.

## Chromatin immunoprecipitation combined with high-throughput sequencing (ChIP-Seq)

Ten-day-old *35S:S-HsfA2-GFP* transgenic *Arabidopsis* plants were treated at 45°C for 2 hr following recovery at 22°C for 20 hr. Two independent transgenic lines (#4 and #6) were generated and separately sequenced for each ChIP-Seq. The plant samples were subjected to ChIP assays via an anti-GFP antibody (ab290, Abcam) according to the instructions for the EpiQuik Plant ChIP Kit (Epigentek). Enriched DNA was used to generate sequencing libraries via the nano-ChIP-Seq protocol as previously described (*Adli and Bernstein, 2011*). The libraries were sequenced via the Illumina HiSeq 2500 platform at Majorbio (Shanghai, China). Following quality control of the resulting sequencing data, MACS software was used to analyse the ChIP-Seq data to obtain the peak (enrichment region) location and information. The S-HsfA2-binding motifs were identified via the MEME-ChIP tool (*Machanick and Bailey, 2011*). Our raw data have been deposited in the NCBI database under BioProject accession number PRJNA947075.

## Protein expression and extraction

Total protein was extracted from the plant samples according to *Liu et al., 2013*.

The GST-tagged proteins (GST–S-HsfA2), GST mock proteins, His6-tagged proteins (His6-HsfA2/S--HsfA2/S-HsfA4c/S-HsfB1/S-HsfB2a/DBD/tDBD), and His6 mock proteins were purified via glutathione Sepharose 4B beads (GE Healthcare) and nickel–nitrotriacetic acid (Ni–NTA) agarose beads (CwBio, China), respectively. Total protein was extracted from the plant samples according to *Liu et al., 2013*.

Twenty-day-old *Arabidopsis* plants were left untreated (control, 22°C) or treated at 37 or 42°C for 2 hr, after which 2 g of each sample was subjected to nuclear protein extraction. Nuclear fractionation was performed according to the protocol described by *Xia et al., 1997*, with modifications to that of *Wang et al., 2020*. Protein concentrations were determined by using a SpectraMax QuickDrop and bovine serum albumin (Sigma) as the reference standards for analysis.

## Y1H and EMSAs

Y1H was performed as described previously (*Sun et al., 2015*).

The 5′ biotin-labelled DNA probes or the corresponding unlabelled DNA were synthesized by Sangon Biotechnology (Shanghai, China). The top strand and the bottom strand were reacted in annealing buffer (50 mM Tris-HCl (pH 7.5), 250 mM NaCl, 0.5 mM EDTA) at a 1:1 ratio. The mixture was heated to 95°C for 4 min to remove all secondary structures and then annealed at 60–70°C for 20 min. Subsequently, the temperature was gradually decreased (such as 5°C/min) to 25°C and maintained for 1–2 hr. EMSA was performed via the LightShift Chemiluminescent EMSA Kit (Pierce) according to the manufacturer's instructions. A total of 100 ng of protein was incubated together with 1.5 ng of biotin-labelled probes in 25 μl reaction mixtures (1× binding buffer, 50 ng

of poly(deoxyinosinic-deoxycytidylic acid), 2.5% (vol/vol) glycerol, 0.05% (vol/vol) Nonidet P-40, and 5 mM $MgCl_2$) for 30 min at room temperature. For the cold competitor, the unlabelled competitors were added to the reaction mixture. The reaction mixtures were separated on 6% (wt/vol) native PAGE gels. The labelled probes were detected according to the instructions provided with the EMSA kit via a Fujifilm LAS-3000 imager.

## GUS staining

The *GUS* reporter transgenic *Arabidopsis* plants were immersed in precooled 90% [vol/vol] acetone for 20 min and then soaked in GUS staining solution (0.5 mg/ml X-Gluc, 100 mM $Na_3PO_4$ [pH = 7.0], 50 µM $K_4[Fe_6]·3H_2O$, 50 µM $K_3[Fe_6]$, 1 mM EDTA [pH = 8.0]) at 37°C in the dark for 12 to –24 hr. After elution with 100% [vol/vol] anhydrous ethanol several times, observations were made with a microscope.

## Western blot analysis

Proteins separated on a gel were electrophoretically transferred to a pure nitrocellulose blotting membrane (Pall Life Sciences). The membrane was cut across the molecular mass region of the corresponding proteins and separately probed with the following corresponding antibodies: an anti-β-glucuronidase (N-terminal) antibody (G5420, Sigma-Aldrich) and an anti-tubulin antibody (T5168, Sigma-Aldrich) for the expression of GUS in *HRE-35Sm:GUS*, *mHRE-35Sm:GUS*, and *35Sm:GUS* seedlings; an anti-mCherry antibody (ab213511, Abcam) for the detection of *35S:HsfA2-RFP* transgenic *Arabidopsis* seedlings; an anti-GST antibody (EASYBIO) and an anti-His6 antibody (Tiangen Biotech, China) for the expression of GST- or His6-tagged proteins; an anti-histone H3 antibody (AS10710, Agrisera, Vännäs, Sweden); and an anti-UGPase antibody (AS05086, Agrisera) to verify successful nuclear protein isolation. Chemiluminescence was performed on a Fujifilm LAS-4000 imager with ECL Prime Western Blot detection reagent (Amersham Biosciences). The ImageJ analysis tool was used for grayscale analysis of the Western blot bands of GUS and tubulin.

## ChIP–qPCR

Ten-day-old *35S:S-HsfA2-Flag* and *35S:HsfA2-RFP* plants were subjected to ChIP assays. The EpiQuik Plant ChIP Kit (Abcam), anti-Flag (F3165-2MG; Sigma), anti-RFP (ab213511; Abcam) and normal mouse IgG (provided by the kit) were used for the assay. After immunoprecipitation, the enriched DNA was analysed via qPCR. Quantification was carried out using the input DNA relative to 1%. The enrichment of the ChIP target was defined as the binding rate between the immunoprecipitated samples of the Flag and RFP antibodies and the control immunoprecipitated sample of IgG. As noted above, the binding ratio was calculated via the triangular incremental period threshold method. ChIP–qPCR for the plant materials described above was performed using the 3′-UTR as a negative control for HRE–HRE-like sequences as well as for the HSE.

## Y2H assay

The bait plasmid BD-S-HsfA2 and a series of AD-HSF prey plasmids were cotransformed into the Y2H Gold Cell yeast strain (Weidi, Shanghai, China). The GAL4 BD bait vector (pGBKT7) and GAL4 AD prey vector (pGAD424) combination were used as negative controls. pGBKT7-53 (BD-53) and pGADT7-T (AD-7) were transformed as positive controls. Their interaction was identified via a spot assay.

## GST pulldown assay

Equal amounts (10 µg) of bacterially expressed and purified proteins, that is, GST mock with His6 mock, GST mock with His6-HsfA2, GST–S-HsfA2 with His6 mock, or GST–S-HsfA2 with His6-HsfA2, were incubated in binding buffer (50 mM Tris-HCl [pH 7.5], 100 mM NaCl, 0.1% Triton X-100 [vol/vol]) overnight in a 4°C rotator (with 50 µl used as the input). Fifty microlitres of GST beads were added, and the mixture was incubated at 4°C for 2 hr. After washing with washing buffer (50 mM Tris-HCl, pH 7.5; 150 mM NaCl; 0.1% Triton X-100 [vol/vol]), the proteins were eluted with 20–30 µl of 20 mM GSH. With input as a positive control, anti-GST and anti-His6 antibodies were subsequently used to detect His6-HsfA2 and GST–S-HsfA2 in the pull-down samples by immunoblotting.

## BiFC assay

Paired nYFP and cYFP constructs, that is, HsfA2-nYFP/S-HsfA2-cYFP, HsfA2ΔN-nYFP/S-HsfA2-cYFP, HsfA2ΔNΔDBD-nYFP/S-HsfA2-cYFP, GUS-nYFP (nYFP negative control)/S-HsfA2-cYFP, HsfA2-nYFP/GUS-cYFP (cYFP negative control), and GUS-nYFP/GUS-cYFP, were coinfiltrated into *N. benthamiana* leaves for 48 hr. After DAPI staining for 30 min to 1 hr, the YFP fluorescence signal was acquired via confocal microscopy (Axio Imager M2; Germany) under the YFP and DAPI channels.

## DNA−protein pulldown assay

DNA pulldown experiments were performed via Pulldown Kits for Biotin-Probes (Viagene Biotech, Changzhou, China) according to the manufacturer's instructions. Briefly, equal amounts (50 μg) of GST–S-HsfA2 or His6-HsfA2 or His6-HsfA2 were incubated in binding buffer (50 mM Tris-HCl [pH 7.5], 100 mM NaCl, 0.1% Triton X-100 [vol/vol]) in a 4°C rotator for 0 hr (control), 4 hr, or 12 hr. The corresponding samples were incubated with biotin-labelled HSE (7 ng) and 5 ng of polydeoxyinosinic-deoxycytidylic acid (dI-dC) for 40 min at 25°C. Then, the DNA/protein-binding mixture was bound to streptavidin−agarose beads by incubation for 40 min at 25°C with gentle shaking and then washed three times to remove the unbound DNA/proteins. The bead-bound proteins were dissociated by boiling for 5 min in dissociation solution and analysed by immunoblotting with an anti-His6 antibody.

## Dual-luciferase reporter assay

Leaves of *N. benthamiana* were infiltrated with a mixed bacterial mixture of *Agrobacterium tumefaciens* (with each construct at $OD_{600}$ = 0.5) for induction via *sHSP17.6Bp:LUC* together with the paired effectors GUS-cYFP/HsfA2-nYFP, S-HsfA2-cYFP/HsfA2-nYFP, S-HsfA2-cYFP/GUS-nYFP, and GUS-cYFP/GUS-nYFP (vector control). LUC bioluminescence was detected after 48 hr using D-luciferin, sodium salt (GOLDBIO) and a CCD camera (Tanon, China). LUC and REN activities were measured on an automated microplate reader (Varioskan Flash, Thermo, USA) via the Dual Luciferase Reporter Gene Assay Kit (Yeasen Biotechnology, Shanghai, China). The LUC/REN ratio of the GUS control (GUS-cYFP/GUS-nYFP) transformed with *sHSP17.6Bp:LUC* was used as the calibrator (set as 1). Three independent experiments were performed.

## Statistical analysis

Statistical Product and Service Solutions (SPSS23) and Microsoft Excel 2007 (Microsoft Corp) were used to perform one-way ANOVA, and Student's *t* tests were performed for $p < 0.05$.

## Accession numbers

The *Arabidopsis* Genome Initiative numbers for the genes mentioned in this article are as follows: *HsfA2* (AT2G26150), *HsfA4c* (AT5G45710), *HSP17.6B* (AT2G29500), *YLS8* (AT5G08290), *HsfB1* (AT4G36990), *HsfB2a* (AT5G62020), and *EF-1α* (AT1G18070). The sequence data supporting the findings of this study have been deposited in the NCBI database under BioProject accession numbers PRJNA1268688 and PRJNA947075.

# Acknowledgements

We thank Prof. Jing Li (Capital Normal University, China) for critical reading and valuable suggestions. This work was supported by the National Natural Science Foundation of China (grant nos. 32070546 and 31771361 to XQ).

# Additional information

## Funding

| Funder | Grant reference number | Author |
| --- | --- | --- |
| National Natural Science Foundation of China | 32070546 | Xiaoting Qi |

| Funder | Grant reference number | Author |
|---|---|---|
| National Natural Science Foundation of China | 31771361 | Xiaoting Qi |

The funders had no role in study design, data collection, and interpretation, or the decision to submit the work for publication.

## Author contributions

Wanxia Chen, Conceptualization, Data curation, Formal analysis, Methodology, Writing – original draft, Writing – review and editing; Jiaqi Zhao, Zhanxia Tao, Shan Zhang, Xiujuan Bei, Wen Lu, Data curation, Formal analysis, Methodology; Xiaoting Qi, Conceptualization, Data curation, Formal analysis, Supervision, Funding acquisition, Validation, Investigation, Methodology, Writing – original draft, Project administration, Writing – review and editing

## Author ORCIDs

Xiaoting Qi ⓘD https://orcid.org/0000-0001-7172-2391

Reviewer #2 (Public review): https://doi.org/10.7554/eLife.99937.3.sa1
Author response https://doi.org/10.7554/eLife.99937.3.sa2

# Additional files

## Supplementary files

Supplementary file 1. Heat stress-responsive genes (HRGs) in S-HsfA2-OE.

Supplementary file 2. Heat stress-responsive genes (HRGs) in S-HsfA2$^{L-A}$-OE.

Supplementary file 3. The shared heat stress-responsive genes (HRGs; including differentially regulated HRGs).

Supplementary file 4. Gene Ontology (GO) enrichment analysis of 848 shared heat stress-responsive genes (HRGs).

Supplementary file 5. A total of 80 putative targets of S-HsfA2.

Supplementary file 6. A list of primers used in this study.

MDAR checklist

## Data availability

DNA and RNA sequence data were deposited in the NCBI database under BioProject accession numbers PRJNA1268688 and PRJNA947075, respectively. All data generated or analysed during this study are included in the manuscript and supporting files; source data files have been provided for all figures.

The following datasets were generated:

| Author(s) | Year | Dataset title | Dataset URL | Database and Identifier |
|---|---|---|---|---|
| Chen W, Zhao J, Tao Z, Zhang S, Bei X, Lu W, Qi X | 2025 | Transcriptome analysis of S-HSF overexpression under heat treatment | https://www.ncbi.nlm.nih.gov/bioproject/?term=PRJNA1268688 | NCBI BioProject, PRJNA1268688 |
| Chen W, Zhao J, Tao Z, Zhang S, Bei X, Lu W, Qi X | 2023 | ChIP-Seq of transgenic *Arabidopsis thaliana* | https://www.ncbi.nlm.nih.gov/search/all/?term=PRJNA947075 | NCBI BioProject, PRJNA947075 |

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
