## [Editor Report · eLife Assessment]

The paper reports **valuable** findings about the mechanism of regulation of the heat shock response in plants that acts as a brake to prevent hyperactivation of the stress response, which have theoretical or practical implications for a subfield. The study presented by the authors provides **solid** methods, data, and analysis that broadly support the claims. This report presents helpful information regarding new spliced HSFs forms in Arabidopsis that highlights key information in the understanding of heat stress and plant growth.

---

## [Referee Report · Reviewer #2 (Public review)]

Summary:

The authors report that Arabidopsis short HSFs S-HsfA2, S-HsfA4c, and S-HsfB1 confer extreme heat. They have truncated DNA binding domains that bind to a new heat-regulated element. Considering Short HSFA2, the authors have highlighted the molecular mechanism by which S-HSFs prevent HSR hyperactivation via negative regulation of HSP17.6B. The S-HsfA2 protein binds to the DNA binding domain of HsfA2, thus preventing its binding to HSEs, eventually attenuating HsfA2-activated HSP17.6B promoter activity. This report adds insights to our understanding of heat tolerance and plant growth.

Strengths:

(1) The manuscript represents ample experiments to support the claim.

(2) The manuscript covers a robust number of experiments and provides specific figures and graphs to in support of their claim.

(3) The authors have chosen a topic to focus on stress tolerance in changing environment.

(4) The authors have summarized the probable mechanism using a figure.

Weaknesses:

Quite minimum

(1) Fig. 3. the EMSA to reveal binding

(2) Alignment of supplementary figures 6-7.

---

## [Author Response]

The following is the authors’ response to the original reviews.

**Reviewer #1 (Public review):**
In the present work, Chen et al. investigate the role of short heat shock factors (S-HSF), generated through alternative splicing, in the regulation of the heat shock response (HSR). The authors focus on S-HsfA2, an HSFA2 splice variant containing a truncated DNA-binding domain (tDBD) and a known transcriptional-repressor leucin-rich domain (LRD). The authors found a two-fold effect of S-HsfA2 on gene expression. On the one hand, the specific binding of S-HsfA2 to the heat-regulated element (HRE), a novel type of heat shock element (HSE), represses gene expression. This mechanism was also shown for other S-HSFs, including HsfA4c and HsfB1. On the other hand, S-HsfA2 is shown to interact with the canonical HsfA2, as well as with a handful of other HSFs, and this interaction prevents HsfA2 from activating gene expression. The authors also identified potental S-HsfA2 targets and selected one, HSP17.6B, to investigate the role of the truncated HSF in the HSR. They conclude that S-HsfA2-mediated transcriptional repression of HSP17.6B helps avoid hyperactivation of the HSR by counteracting the action of the canonical HsfA2.The manuscript is well written and the reported findings are, overall, solid. The described results are likely to open new avenues in the plant stress research field, as several new molecular players are identified. Chen et al. use a combination of appropriate approaches to address the scientific questions posed. However, in some cases, the data are inadequately presented or insufficient to fully support the claims made. As such, the manuscript would highly benefit from tackling the following issues:(1) While the authors report the survival phenotypes of several independent lines, thereby strengthening the conclusions drawn, they do not specify whether the presented percentages are averages of multiple replicates or if they correspond to a single repetition. The number of times the experiment was repeated should be reported. In addition, Figure 7c lacks the quantification of the hsp17.6b-1 mutant phenotype, which is the background of the knock-in lines. This is an essential control for this experiment

For the seedling survival rates and gene expression levels, we added statistical analysis based on at least two independent experiments. Figure 6E of the revised manuscript shows the phenotypes of the WT, hsp17.6b-1, HSP17.6B-KI, and HSP17.6B-OE plants and the statistical analysis of their seedling survival rates after heat exposure.

(2) In Figure 1c, the transcript levels of HsfA2 splice variants are not evident, as the authors only show the quantification of the truncated variant. Moreover, similar to the phenotypes discussed above, it is unclear whether the reported values are averages and, if so, what is the error associated with the measurements. This information could explain the differences observed in the rosette phenotypes of the S-HsfA2-KD lines. Similarly, the gene expression quantification presented in Figures 4 and 5, as well as the GUS protein quantification of Figure 3F, also lacks this crucial information.

RT‒qPCR analysis of the expression of these genes from at least two independent experiments was performed. We also added these missing information to the figure legends.

(3) The quality of the main figures is low, which in some cases prevents proper visualization of the data presented. This is particularly critical for the quantification of the phenotypes shown in Figure 1b and for the fluorescence images in Figures 4f and 5b. Also, Figure 9b lacks essential information describing the components of the performed experiments.

We apologize; owing to the limitations of equipment and technology, we will attempt to obtain high-quality images in the future. A detailed description of Figure 9b is provided in the methods section.

(4) Mutants with low levels of S-HsfA2 yield smaller plants than the corresponding wild type. This appears contradictory, given that the proposed role of this truncated HSF is to counteract the growth repression induced by the canonical HSF. What would be a plausible explanation for this observation? Was this phenomenon observed with any of the other tested S-HSFs?

We found that the constitutive expression of S-HsfA2 inhibits Arabidopsis growth. Considering this, Arabidopsis plants do not produce S-HsfA2 under normal conditions to avoid growth inhibition. However, under heat stress, Arabidopsis plants generate S-HsfA2, which contributes to heat tolerance and growth balance. In the revised manuscript, we provided supporting data indicating that S-HsfA4c-GFP or S-HsfB1-RFP constitutive expression confers Arabidopsis extreme heat stress sensitivity but inhibits root growth (Supplemental Figure S8). Therefore, this phenomenon is also observed in S-HsfA4c-GFP or S-HsfB1-RFP.

(5) In some cases, the authors make statements that are not supported by the results:(i) the claim that only the truncated variant expression is changed in the knock-down lines is not supported by Figure 1c;

In three S-HsfA2-KD lines, RT‒PCR splicing analysis revealed that HsfA2-II but not HsfA2-III is easily detected. In the revised manuscript, we added RT‒qPCR analysis, and the results revealed that the abundance of HsfA2-III and HsfA2-II but not that of the full-length HsfA2 mRNA significantly decreased under extreme heat (Figure 1C). Considering that HsfA2-III but not HsfA2-II is a predominant splice variant under extreme heat (Liu et al., 2013), S-HsfA2-KD may lead to the knockdown of alternative HsfA2 splicing transcripts, especially HsfA2-III.

(ii) the increase in GUS signal in Figure 3a could also result from local protein production;

We included this possibility in the results analysis.

(iii) in Figure 6b, the deletion of the HRE abolishes heat responsiveness, rather than merely altering the level of response; and

In the revised manuscript, we added new data concerning the roles of HREs and HSEs in the response of the HSP17.6B promoter to heat stress (Figure 6A). These results suggest that the HRE and HSE elements are responsible for the response of the HSP17.6B promoter to heat stress and that the HRE negatively regulates the HSP17.6B promoter at 37°C, whereas the HSE is positive at 42°C.

(iv) the phenotypes in Figure 8b are not clear enough to conclude that HSP17.6B overexpressors exhibit a dwarf but heat-tolerant phenotype.

When grown in soil, the HSP17.6B-OE seedlings presented a dwarf phenotype compared with the WT control. Heat stress resulted in browning of the WT leaves, but the leaves of the HSP17.6B-OE plants remained green, suggesting that the HSP17.6B-OE seedlings also presented a heat-tolerant phenotype in the soil. These results are qualitative but not quantitative experimental data; therefore, the conclusions are adjusted in the abstract and results sections.

**Reviewer #2 (Public review):**
Summary:The authors report that Arabidopsis short HSFs S-HsfA2, S-HsfA4c, and S-HsfB1 confer extreme heat. They have truncated DNA binding domains that bind to a new heat-regulated element. Considering Short HSFA2, the authors have highlighted the molecular mechanism by which S-HSFs prevent HSR hyperactivation via negative regulation of HSP17.6B. The S-HsfA2 protein binds to the DNA binding domain of HsfA2, thus preventing its binding to HSEs, eventually attenuating HsfA2-activated HSP17.6B promoter activity. This report adds insights to our understanding of heat tolerance and plant growth.Strengths:(1) The manuscript represents ample experiments to support the claim.(2) The manuscript covers a robust number of experiments and provides specific figures and graphs in support of their claim.(3) The authors have chosen a topic to focus on stress tolerance in a changing environment.Weaknesses:(1) One s-HsfA2 represents all the other s-Hsfs; S-HsfA4c, and S-HsfB1. s-Hsfs can be functionally different. Regulation may be positive or negative. Maybe the other s-hsfs may positively regulate for height and be suppressed by the activity of other s-hsfs.

In this study, we used S-HsfA2, S-HsfA4c, and S-HsfB1 data to support the view that “splice variants of HSFs generate new plant HSFs”. We also noted that S-HsfA2 cannot represent a traditional S-HSF. S-HsfA4c and S-HsfB1 may have functions other than S-HsfA2 because of their different C-terminal motifs or domains. Different S-HSFs might participate in the same biological process, such as heat tolerance, through the coregulation of downstream genes. We added this information to the discussion section.

(2) Previous reports on gene regulations by hsfs can highlight the mechanism.

In the introduction section, we included these references concerning HSFs and S-HSFs.

(3) The Materials and Methods section could be rearranged so that it is based on the correct flow of the procedure performed by the authors.

The materials and methods and results sections are arranged in the logical order.

(4) Graphical representation could explain the days after sowing data, to provide information regarding plant growth.

The days after sowing (DAS) for the age of the Arabidopsis seedlings are stated in the Materials and Methods section and figure legends.

(5) Clear images concerning GFP and RFP data could be used.

We provided high-quality images of S-HsfA2-GFP and the GFP control (Figure 3 in the revised manuscript).

**Reviewing Editor comments:**
The EMSA shown in Figures 2, 3, 4, and 5, which are critical to support the manuscript's claims, are of poor quality, without any repeats to support. In addition, there is not much information about how these EMSA were done. I suggest including better EMSA in a new version of this manuscript.

Thank you for your suggestion. We added the missing information, including the detailed EMSA method and experiment repeat times in the methods section and figure legends. We provide high-quality images of HRE probes binding to nuclear proteins (Figure 4E).

**Reviewer #1 (Recommendations for the authors):**
(1) The paper is overall well-written, but it could greatly benefit from reorganizing the results subsections. Currently, there are entire subsections dedicated to supplementary figures (e.g., lines 177-191) and main figures split into different subsections (e.g., lines 237-246). It is recommended to organize all the information related to a main figure into a single subsection and to incorporate the description of the corresponding supplementary figures. This would imply a general reorganization of the figures, moving some information to the supplementary data (for instance, the data in Figure 4 could be supplementary to Figure 5) and vice versa (Supplementary Figure 4 should be incorporated into main Figure 2, as it presents very important results). Also, Figures 7 and 8 would be better presented if merged into a single figure/subsection.

Thank you for your suggestion. We have merged some figures into a single figure according to the main information. In the current version, there are 8 main figures, which includes a new figure.

(2) Survival phenotypes vary widely, making reliable statistical analysis challenging. The chlorophyll and fresh weight quantifications presented in figures such as Figure 5 appear to effectively describe the phenomenon and allow for statistical comparisons. Figures 1 and 7 would benefit from including these measurements if the variability in survival percentages is too high to calculate statistical differences reliably. Also, in Figure 8, all chlorophyll measurements should be normalized to fresh weight rather than seedling number due to the dwarfism observed in the overexpressor lines.

Thank you for pointing out your concerns. We added statistical analysis based on at least two independent experiments, including Figures 1 and 7, to the original manuscript. In Figure 8 in the original manuscript, chlorophyll measurements were normalized to fresh weight.

(3) Typos: in Figure 3a it should be "min" not "mim"; in Supplementary Figure 3, the GFP and merge images are swapped.

We apologize for these errors, and we have corrected them. Supplementary Figure 3 was replaced with new images and was included in Figure 3 in the revised manuscript.

**Reviewer #2 (Recommendations for the authors):**
(1) The abstract states "How this process is prevented to ensure proper plant growth has not been determined." The authors can be the first to do this, by adding graphical data on the height difference in hSfA2-arabidopis and wild-type Arabidopsis.

Thank you and agree with you. We have added this information to the new working model (Figure 8)

(2) The authors claim that Arabidopsis S-HsfA2, S-HsfA4c, and S-HsfB1; but have used S-HsfA2 to understand the action. The mechanisms being unknown for S-HsfA4c, and S-HsfB1 cannot be represented by S-HsfA2 to represent the mechanism.

Thank you for your valuable comments. In this study, we used S-HsfA2, S-HsfA4c, and S-HsfB1 data to support the view that “splice variants of HSFs generate new plant HSFs”. We also noted that S-HsfA2 cannot represent a traditional S-HSF because S-HsfA4c and S-HsfB1 may have functions other than S-HsfA2. Therefore, we deleted “representative S-HSF” from the revised manuscript. In the future, we will conduct in-depth research on the relevant mechanisms of S-HsfA4c and S-HsfB1 under your guidance.

(3) The authors can include which of the HSFs interacted with other genes of Arabidopsis reported by other researchers are positively or negatively regulated in heat response/ growth or the balance.

In the introduction section, we included these genes. AtHsfA2, AtHsfA3, and BhHsf1 confer heat tolerance in Arabidopsis but also result in a dwarf phenotype in plants (Ogawa et al., 2007; Yoshida et al., 2008; Zhu et al., 2009).

(4) The authors have started from the subsection plant materials and growth conditions. It is unclear from where the authors have found these HSF mutant Arabidopsis? Is it a continuation of some other work? As a reader, I am utterly confused because of the arrangement of the materials and methods section.

We apologize for the lack of detailed information in the Materials and methods section. These mutants were purchased from AraShare (Fuzhou, China) and verified via PCR and RT‒qPCR. We added the missing information.

(5) Is the DAS - Days After Sowing - represented as a graph or table? This will add data to the plant growth section to clearly state the difference between the mutants and the wild-type.

In this study, the age of the Arabidopsis seedlings was calculated as days after sowing (DAS), as stated in the Materials and Methods section and figure legends.

(6) Heat stress treatment after gus staining looks absurd. Should it not follow after plant materials and growth conditions, which should ideally be after the plant transformation and cloning section? The initial step is definitely about plasmid construction. Kindly rearrange.

Thank you for your valuable suggestions. We have rearranged the logical order of the materials and methods.

(7) The expression of GFP and RFP was not clearly seen in the images. This could be because of the poor resolution of the images added.

We obtained high-quality images of S-HsfA2-GFP (Figure 3 in the revised manuscript).

(8) We live in an age where it is widely known that genes are not functioning independently but are coregulated and coregulate other proteins. The authors can address the role of these spliced variants on gene regulation and compare them with the HSFs.

We agree with your suggestion. In this study, HSP17.6B was identified as a direct gene of S-HsfA2 and HsfA2, which can partly explain the role of S-HsfA2 in heat resistance and growth balance. However, the mechanical mechanism by which S-HsfA2 regulates heat tolerance and growth balance may not be limited to HSP17.6B. On the basis of the current data, we propose that the putative S-HsfA2-DERB2A-HsfA3 module might be associated with the roles of S-HsfA2 in heat tolerance and growth balance. Please refer to the discussion section for a detailed explanation.

(9) Regulatory elements can be validated in relation to their interaction with proven HSFs.

Supplemental Figure S3 shows that His6-HsfA2 failed to bind to the HRE in vitro.

(10) The authors seem to be biased toward heat stress and have not worked enough on plant growth. Biochemical data and images on plant growth could be added to bring out the novelty of this manuscript.

Thank you for your suggestion. We added new data indicating that, compared with the wild-type control, S-HsfA2-GFP, S-HsfA4c-GFP, or S-HsfB1-GFP overexpression inhibited root length (Supplemental Figure 8).

(11) Line 251 on page 11 of the submitted manuscript says that the s-Hsfs were previously identified by Liu et al. (2013) yet in the abstract the authors claim that these s-HsFs are NEW kinds of HSF with a unique truncated DNA-binding domain (tDBD) that binds a NEW heat-regulated element (HRE).

In our previous report, several S-HSFs, including S-HsfA2, S-HsfA4, S-HsfB1, and S-HsfB2a, were identified primarily in Arabidopsis (Liu et al., 2013). In this study, we further characterized S-HsfA2, S-HsfA4, and S-HsfB1 and revealed several features of S-HSFs. Therefore, we claim that these S-HSFs are new kinds of HSFs.

(12) What are these NEW kinds of HRE? Which genes have these HRE? Was an in silico study conducted to study it or can any reports can be cited?

HREs, i.e., heat-regulated elements, are newly identified heat-responsive elements in this study. The sequences of HREs are partially related to traditional heat shock elements (HSEs). Because we did not identify the essential nucleic acids required for t-DBD binding to the HRE, we did not perform an in silico study.

(13) S-HSFs may interact with existing HSFs. Have the authors thought in this direction? It can have a role in positively regulating other sHSFs or regulating multiple expressing genes related to plant growth and other functions. This needs to be explored.

Thank you for this point. Given that the overexpression of Arabidopsis HsfA2 or HsfA3 inhibits growth under nonstress conditions, we discussed this direction from the perspective of the interaction of S-HsfA2 with HsfA2 or HsfA3 in the revised manuscript.

(14) The authors need to concentrate on the presentation and arrangement of both their materials and methods and result section and write them in a systematic manner (or following a workflow).

The materials, methods and results sections are arranged in logical order.

(15) The authors have used references in the results section which can be added to the discussion section to make it more accurate.

Thank you for your suggestions. We have moved some references to the discussion section, but the necessary references remain in the results section.